# Glycolysis regulates Hedgehog signalling via the plasma membrane potential

Stephanie Spannl[1,‡,*] , Tomasz Buhl[1,2,¶], Ioannis Nellas[1,2,¶], Salma A Zeidan[1,2], K Venkatesan Iyer[1,3], Helena Khaliullina[1,§], Carsten Schultz[4,5] , André Nadler[1], Natalie A Dye[1,**] & Suzanne Eaton[1,2,6,†]

## Abstract

Changes in cell metabolism and plasma membrane potential have been linked to shifts between tissue growth and differentiation, and to developmental patterning. How such changes mediate these effects is poorly understood. Here, we use the developing wing of *Drosophila* to investigate the interplay between cell metabolism and a key developmental regulator—the Hedgehog (Hh) signalling pathway. We show that reducing glycolysis both lowers steady-state levels of ATP and stabilizes Smoothened (Smo), the 7-pass transmembrane protein that transduces the Hh signal. As a result, the transcription factor Cubitus interruptus accumulates in its full-length, transcription activating form. We show that glycolysis is required to maintain the plasma membrane potential and that plasma membrane depolarization blocks cellular uptake of N-acylethanolamides—lipoprotein-borne Hh pathway inhibitors required for Smo destabilization. Similarly, pharmacological inhibition of glycolysis in mammalian cells induces ciliary translocation of Smo—a key step in pathway activation—in the absence of Hh. Thus, changes in cell metabolism alter Hh signalling through their effects on plasma membrane potential.

**Keywords** endocannabinoids; glycolysis; hedgehog signalling; metabolism; plasma membrane potential

**Subject Categories** Development; Membranes & Trafficking; Metabolism

**The EMBO Journal (2020) 39: e101767**

See also: **Q Dong & LY Cheng** (November 2020)

## Introduction

Tissue growth is associated with changes in cell metabolism that limit the complete catabolism of glucose to $CO_2$, diverting it towards the synthesis of cellular building blocks (Warburg, 1956; Cairns *et al*, 2011; Agathocleous & Harris, 2013). The metabolic state of the cell can also influence gene expression and help specify differentiation programs. For example, changes in the level of tricarboxylic acid (TCA) cycle metabolites, such as acetate or α-ketoglutarate, can alter histone acetylation and methylation (Kaelin & McKnight, 2013). Less is known about how cell metabolism might affect the activity of developmental signalling pathways that control growth and patterning.

One conserved signalling system that controls growth and patterning from arthropods to vertebrates is the Hedgehog (Hh) pathway (Ingham *et al*, 2011). Here, we probe the effects of cell metabolism on the cell biological mechanisms underlying Hh signalling in the *Drosophila* wing imaginal disc. In the wing disc, Hh is produced in the posterior compartment, while its receptor Patched (Ptc) is expressed in the anterior compartment. In cells that are not exposed to Hh, Ptc destabilizes a key transducer of the Hh signal—the 7-pass transmembrane protein Smoothened (Smo), reducing its levels on the basolateral membrane. Ptc is a $Na^+$-dependent RND family transporter that is thought to inhibit Smo by modulating the trafficking of small lipidic Smo regulators. Smo can be activated by cholesterol binding in vertebrates (Byrne *et al*, 2016; Huang *et al*, 2016; Luchetti *et al*, 2016; Myers *et al*, 2017; Xiao *et al*, 2017) or inhibited by molecules such as N-acylethanolamines in both vertebrates and *Drosophila* (Khaliullina *et al*, 2015). As Hh spreads into the anterior compartment of the wing disc, it inhibits Ptc activity, thereby stabilizing Smo on the plasma membrane near the anterior–posterior (A/P) compartment boundary. Smo signalling blocks proteasomal processing of Cubitus interruptus (Ci), a Gli-family transcription factor, and changes its activity from a transcriptional repressor ($Ci_{75}$) to a transcriptional activator ($Ci_{155}$). $Ci_{155}$ then activates transcription of different target genes at different distances from the A/P boundary,

1 Max Planck Institute of Molecular Cell Biology and Genetics, Dresden, Germany
2 Biotechnologisches Zentrum, Technische Universität Dresden, Dresden, Germany
3 Max Planck Institute for the Physics of Complex Systems, Dresden, Germany
4 Cell Biology and Biophysics Unit, European Molecular Biology Laboratory, Heidelberg, Germany
5 Department of Chemical Physiology and Biochemistry, Oregon Health and Science University, Portland, OR, USA
6 Center for Systems Biology Dresden, Dresden, Germany
  *Corresponding author. Tel: +1 416 946 3820; E-mail: steffi.spannl@utoronto.ca
  **Corresponding author. Tel: +49 351 210 2806; E-mail: dye@mpi-cbg.de
  ¶These authors contributed equally to this work
  †Deceased July 2, 2019
  ‡Present address: Department of Biochemistry, Faculty of Medicine, University of Toronto, Toronto, ON, Canada
  §Present address: Department of Physiology, Development and Neuroscience, University of Cambridge, Cambridge, UK

including *decapentaplegic* (*dpp*), *engrailed* (*en*) and *ptc*. Upregulation of *ptc* transcription by Hh signalling is thought to limit the spread of the Hh ligand (Briscoe & Therond, 2013).

Interestingly, imaginal wing disc cells can release Hh in several different forms. One form is covalently modified by sterol at the C-terminus and a fatty acid at the N-terminus (Porter *et al*, 1996a,b; Pepinsky *et al*, 1998; Chamoun *et al*, 2001; Lee & Treisman, 2001; Micchelli *et al*, 2002). This form of Hh can be secreted on lipoproteins derived from the circulation, released on exosomes, or spread on membrane protrusions (Panakova *et al*, 2005; Bischoff *et al*, 2013; Gradilla *et al*, 2014; Matusek *et al*, 2014). In addition, wing discs release Hh in a monomeric, non-sterol modified form (HhN) (Palm *et al*, 2013). Lipoprotein-associated (Lpp-associated) Hh can stabilize Smo and cause accumulation of full-length $Ci_{155}$ by preventing lipoprotein-derived N-acylethanolamides from destabilizing Smo (Khaliullina *et al*, 2009, 2015; Palm *et al*, 2013). Lipoprotein-associated Hh cannot activate target gene transcription by itself, but it sensitizes imaginal disc cells to the sterol-free form of Hh (HhN). Together, these two forms activate $Ci_{155}$-dependent target gene activation and growth (Palm *et al*, 2013).

Research on the protein Ecdysoneless (Ecd) has provided a possible connection between Hh signalling and cell metabolism. In *Drosophila*, Ecd is an interacting partner of the core splicing machinery that regulates steroid hormone production in the ring gland (Claudius *et al*, 2014). It is also required for imaginal tissue growth, independent of steroid production (Redfern & Bownes, 1983; Sliter, 1989; Gaziova *et al*, 2004). A connection between Ecd activity and Hh signalling was suggested from early experiments showing that *ecd* mutants could be partially rescued by driving Ecd expression in cells that receive the Hh signal but not in those that produce it (Gaziova *et al*, 2004). Subsequently, it has been shown that Ecd promotes glycolysis in both yeast and human tumours (Kainou *et al*, 2006; Dey *et al*, 2012). Given this function for Ecd in other systems, the genetic rescue experiment in *Drosophila* raises the possibility that glycolysis is important in Hh-receiving cells for signal transduction.

Here, we address the open question of how cell metabolism influences developmental patterning systems by studying the interplay between glycolysis and the Hh signalling pathway. Upon downregulating glycolytic enzymes and Ecd in the *Drosophila* wing disc, and pharmacologically perturbing glycolytic activity in mammalian cell culture, we find an upregulation of Hh signalling. We investigate the molecular mechanism in the *Drosophila* wing disc and find that perturbing glycolysis blocks the cellular uptake of N-acylethanolamides by depolarizing the plasma membrane. This change in plasma membrane potential interferes with Smo destabilization and sensitizes receiving cells to the Hh ligand. This work thereby provides a molecular mechanism for how cellular metabolism regulates the activity of a conserved developmental signalling pathway that controls tissue growth and patterning.

# Results

### Reducing glycolytic enzyme expression lowers steady-state levels of ATP and growth

The *Drosophila* wing disc is a classic model system for studying the Hh pathway, but little is known about its metabolism or the importance of glycolysis to its normal developmental growth and signalling. To analyse the effect of perturbing glycolysis on energy homeostasis in the developing *Drosophila* wing, we generated flies expressing a FRET-based reporter of ATP concentration (Tsuyama *et al*, 2013) under the control of the ubiquitously active *ubiquitin* promoter (*ubi-AT1.03NL*) (Fig 1A). The FRET signal generated by this construct is strongly reduced by treatment of explanted wing discs with antimycin A, an inhibitor of oxidative phosphorylation, confirming that the construct is sensitive to cellular ATP levels (Fig 1B and D–F).

---

**Figure 1. Loss of glycolytic enzymes or Ecdysoneless lowers ATP levels.**

A   Schematic illustration of the working principle of the ATP sensor *AT1.03NL*. Binding of ATP to the sensing domain (ε subunit of ATP synthase) leads to a conformational change that brings the donor (mse-CFP) and the acceptor (cp173-Venus) closer, resulting in an increase in FRET efficiency (Tsuyama *et al*, 2013).

B   Cartoon to illustrate the experimental setup for inhibiting oxidative phosphorylation (OxPhos) in late third-instar wing discs expressing *AT1.03NL*. To inhibit OxPhos, dissected wing discs are treated with antimycin A for 2 h. A reduction in ATP is revealed by a reduced FRET efficiency. The grey frame marks the region of the wing disc shown in FRET efficiency and immunofluorescence (IF) images.

C   Cartoon of late third-instar wing disc showing the expression pattern of *ap-GAL4, tub-GAL80^{ts}* (*apGal^{ts}*) driver. Dashed yellow and green lines indicate the anterior–posterior (A/P) and dorsal–ventral (D/V) compartment boundaries, respectively. The expression pattern of *apGal^{ts}* driver in the dorsal compartment is shown in light red.

D, E   Images of FRET efficiency in mock-treated (− ant.A, D) and antimycin A-treated (+ ant. A., E) *ubi-AT1.03NL* wing discs at time point 2 h. A rainbow colormap is used to indicate the FRET efficiency levels. Scale bars = 50 μm.

F   Line graphs showing the FRET efficiency in individual mock-treated (− ant. A, *n* = 26) and ant. A-treated (+ ant. A., *n* = 19) *ubi-AT1.03NL* wing discs before (0 h) and after 2 h of treatment. Paired *t*-test, ns = not significant, ***$P \leq 0.001$.

G–J   Time-controlled knock-down of metabolic enzymes in the dorsal compartment of *ubi-AT1.03NL* wing discs using the *apGal^{ts}* driver. Images of FRET efficiency in control (G), *apGal^{ts}>Pfk^{RNAi}* (H), *apGal^{ts}>Gapdh^{RNAi}* (I) and *apGal^{ts}>Glo1^{RNAi}* (J) wing discs after RNAi induction for 48 h (G), 120 h (H, J) and 93 h (I). A rainbow colormap is used to indicate the FRET efficiency levels. Scale bars = 50 μm.

K   Line graphs showing the FRET efficiency in the ventral and dorsal compartments of individual control (*n* = 27), *apGal^{ts}>Pfk^{RNAi}* (*n* = 7), *apGal^{ts}>Gapdh^{RNAi}* (*n* = 10) and *apGal^{ts}>Glo1^{RNAi}* (*n* = 6) wing discs. Loss of Pfk, Gapdh or Glo1 in the dorsal compartment of the wing disc reduces the levels of ATP. Paired *t*-test, ns = not significant, **$P \leq 0.01$, ***$P \leq 0.001$.

L   Time-controlled knock-down of *ecdysoneless* (*ecd*) in the dorsal compartment of *ubi-AT1.03NL* wing discs using *apGal^{ts}>ecd^{RNAi}*. Image of FRET efficiency in *apGal^{ts}>ecd^{RNAi}* wing disc 48 h after RNAi induction. A rainbow colormap is used to indicate the FRET efficiency levels. Scale bars = 50 μm.

M   Line graph showing the FRET in the dorsal and ventral compartments of *apGal^{ts}>ecd^{RNAi}* (*n* = 28) wing discs. Loss of Ecd in the dorsal compartment of the wing disc reduces the levels of ATP. Paired *t*-test, ***$P \leq 0.001$.

Source data are available online for this figure.

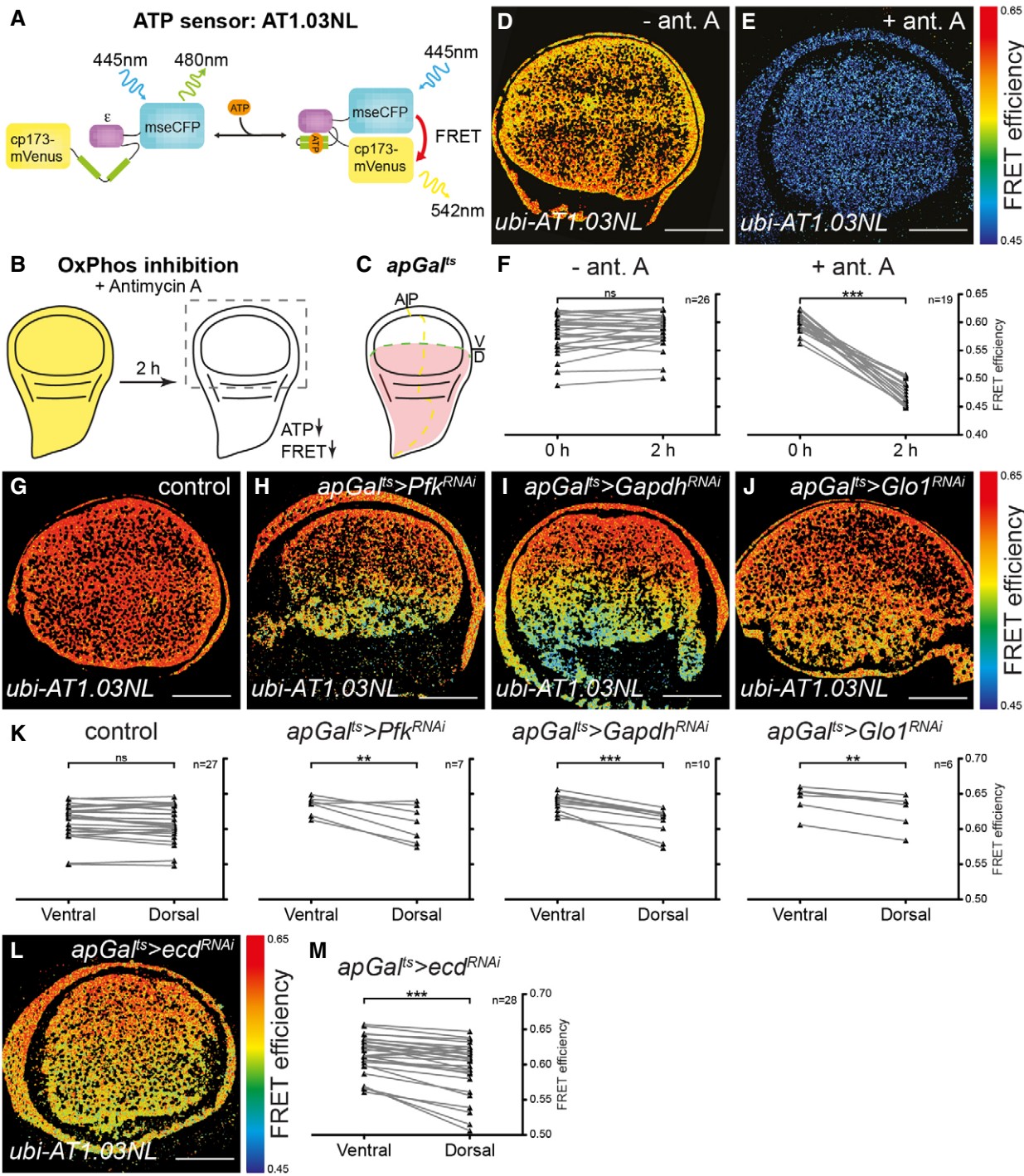

**Figure 1.**

We then used this construct to determine the effect of downregulating key enzymes of the glycolytic pathway on steady-state levels of ATP (Fig 1G–K). We spatially and temporally limited the induction of RNAi, using the *ap-Gal4* driver combined with the temperature-sensitive Gal4 repressor, *tub-Gal80^{ts}* (henceforward denoted *apGal^{ts}*). This driver induces expression only in the dorsal compartment (Fig 1C), leaving the ventral compartment as an internal control, and the temporal control provided by *tub-Gal80^{ts}* allowed us to restrict the duration of the knock-down during larval development. We found that knock-down of *Phosphofructokinase* (*Pfk*), a key control enzyme in glycolysis, lowers steady-state levels of ATP (*apGal^{ts} > Pfk^{RNAi}*, Fig 1H and K). We also tested the effect of removing Glyceraldehyde-3-phosphate dehydrogenase (*Gapdh*). The *Drosophila* genome encodes two isoforms of Gapdh that are 97%

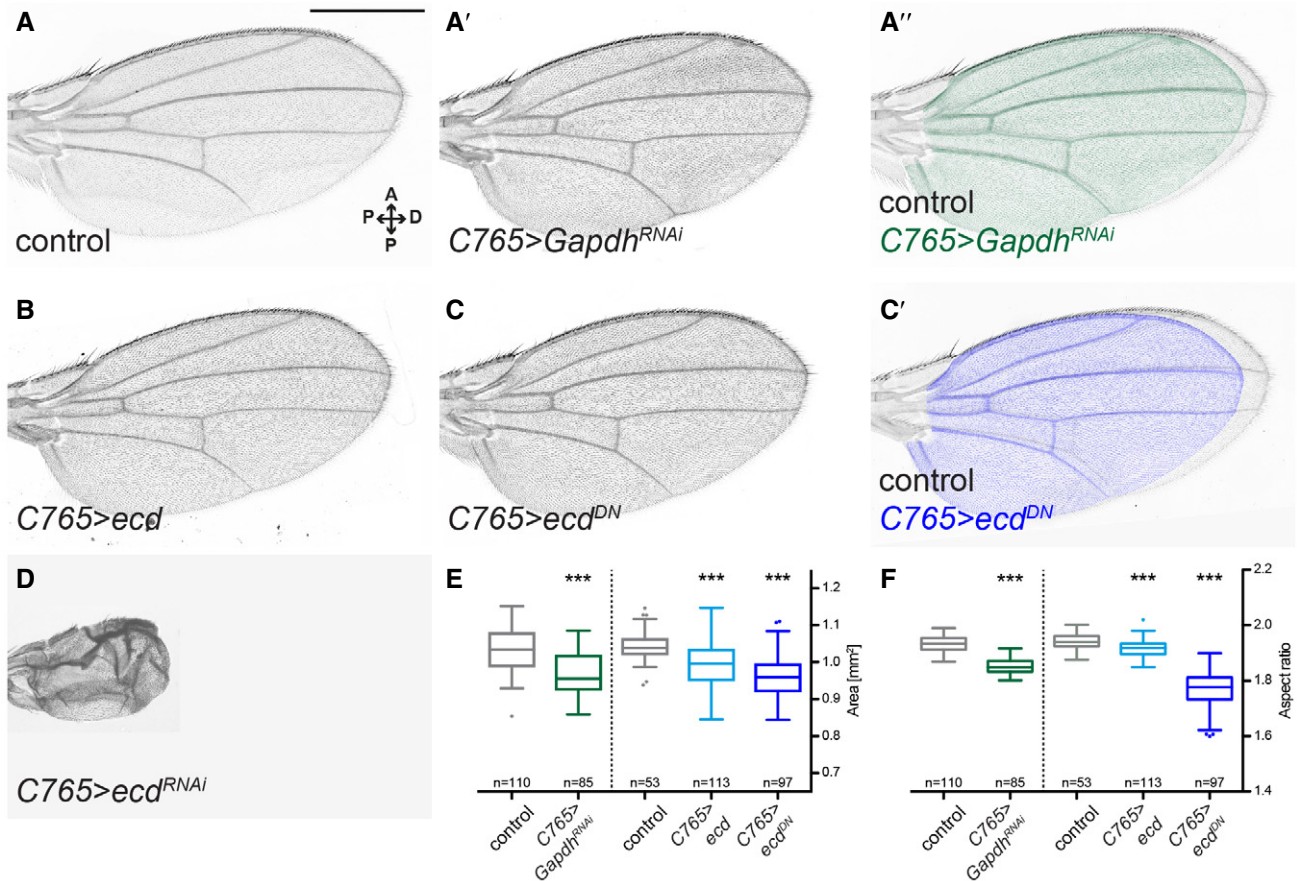

**Figure 2. Knock-down of *Gapdh* and over-expression of dominant-negative Ecd result in smaller, rounder adult wings.**

A–D  Adult wing phenotype of knock-down of *Gapdh* (*C765>Gapdh^RNAi^*), over-expression of full-length Ecd (*C765>ecd*), over-expression of C-terminally truncated form of Ecd (*C765>ecd^DN^*) and knock-down of *ecd* (*C765>ecd^RNAi^*) in the whole wing disc (see Appendix Fig S3A for the expression pattern of *C765-Gal4*). Images of adult wings of control (A), *C765>Gapdh^RNAi^* (A′), *C765>ecd* (B), *C765>ecd^DN^* (C) and *C765>ecd^RNAi^* (D) male flies. (A″) Overlay of the wings shown in A and A′. (C′) Overlay of control wing and wing shown in C and a control wing similar to that shown in A. Loss of Ecd throughout the wing discs produces adults with vestigial wings (D). In contrast, the effects of *ecd^DN^* over-expression or *Gapdh* knock-down are less severe on tissue survival, giving rise to smaller wings with altered proportions. Images consist of a series of tiles that were stitched together by the microscope software. Scale bar = 500 µm.

E, F  Tukey box-and-whiskers plot showing quantifications of the area (E) and shape (aspect ratio: major/minor axis, F) of control, *C765>Gapdh^RNAi^*, *C765>ecd* and *C765>ecd^DN^* wings. Lower and upper hinges correspond to the first and third quartiles; vertical lines extend to ± 1.5 times the interquartile range. Sample size is indicated for each perturbation. Loss of Gapdh or over-expression of Ecd^DN^ leads to smaller wings that are increased along the anterior–posterior (A/P) axis compared to the proximal–distal (P/D) axis. Statistical analysis was performed using one-way ANOVA, followed by Dunnett's multiple comparisons test. ***$P \leq 0.001$.

Source data are available online for this figure.

identical at the protein level and 89% identical at the DNA level (Tso *et al*, 1985; Sun *et al*, 1988). Using an RNAi construct that targets both isoforms (Appendix Fig S1), we found that loss of Gapdh also results in lower steady-state levels of ATP (*apGal^ts^ > Gapdh^RNAi^*, Fig 1I and K). Lastly, we tested the effect of knocking down *Glyoxalase 1* (*Glo1*), which converts methylglyoxal to D-lactate. Methylglyoxal is formed non-enzymatically from the triose phosphate substrates of Gapdh, and it accumulates when either Gapdh or Glo1 enzymatic activity is blocked (Tristan *et al*, 2011; Moraru *et al*, 2018). This highly reactive metabolite can inhibit glycolysis by glyoxylating Gapdh and lactate dehydrogenase (Leoncini *et al*, 1989; Morgan *et al*, 2002; Lee *et al*, 2005). As with *Pfk* and *Gapdh* knock-down, loss of Glo1 during larval development

reduces steady-state levels of ATP, albeit to a lesser degree (*apGal^ts^ > Glo1^RNAi^*, Fig 1J and K).

To assess the functional consequences of reducing glycolysis in the developing wing, we quantified wing size and shape upon knocking down these and other enzymes of glycolysis throughout the entire wing disc during all of the development using *C765-Gal4* (Appendix Fig S2). Some RNAi lines produced wings that were proportionally smaller, while many other RNAi lines produced wings that were slightly misproportioned. This latter group included RNAi lines targeting *Pfk* (1/1 line), *Gapdh2* (2/3 lines), *Glo1* (1/1 line), as well as *Hexokinase A* (1/1 line), *Aldolase* (2/3 lines), *Phosphoglycerate kinase* (*Pgk*) (3/3 lines) and *Pyruvate kinase* (*PyK*) (2/2 lines).

## Ecdysoneless is autonomously required for tissue growth and energy metabolism

Given that Ecd has been shown in yeast and humans to promote glycolysis, we also investigated the possibility that Ecd could be used to more globally alter glycolysis (Kainou *et al*, 2006; Dey *et al*, 2012). A prerequisite for such an approach would be that Ecd indeed acts autonomously and does not influence wing growth indirectly via its function in the steroid-producing gland. To confirm an autonomous role for Ecd in wing disc growth, we performed a spatially and temporally controlled RNAi-mediated knock-down using $apGal^{ts}$ to drive $ecd^{RNAi}$ in the dorsal compartment of the wing disc ($apGal^{ts} > ecd^{RNAi}$). We confirmed that $apGal^{ts}$ does not induce gene expression in the ring gland (Fig EV1A–C), indicating that steroid hormone production should be unperturbed. Induction of $ecd^{RNAi}$ for 48 h depletes a fosmid Ecd::GFP fusion construct (Sarov *et al*, 2016), confirming knock-down efficiency (Fig EV1D and E). Under these conditions, the size of the dorsal compartment is also reduced compared to the ventral compartment (compare Fig EV1D/D′ with E/E′, quantified in G). Staining for activated Caspase-3 (Cas3*) revealed that cell death is only increased in the dorsal compartment after very long inductions of $ecd^{RNAi}$ and not yet by 48 h (Fig 1D″–F″, quantified in H), consistent with what has been observed in *ecd* mutant clones (Gaziova *et al*, 2004; Claudius *et al*, 2014). We conclude that *ecd* knock-down first reduces tissue growth and later activates apoptosis.

We next explored the potential for Ecd to regulate wing disc metabolism. Given that *ecd* knock-down in pancreatic cancer cells lowers steady-state ATP levels (Dey *et al*, 2012), we tested whether its knock-down in *Drosophila* has similar effects. We induced $ecd^{RNAi}$ for 48 h in the dorsal compartment of *ubi-AT1.03NL* wing discs using $apGal^{ts}$ and observed a drop in steady-state levels of ATP in the dorsal $ecd^{RNAi}$ tissue (Fig 1L and M), mimicking the effect of glycolytic enzyme knock-down in the wing (Fig 1G–K) and loss of Ecd in pancreatic cancer cells (Dey *et al*, 2012).

Given that Ecd is known to interact with basic components of the splicing machinery (Guruharsha *et al*, 2011; Havugimana *et al*, 2012; Claudius *et al*, 2014; Hein *et al*, 2015), we investigated the possibility that the reduction of ATP levels upon *ecd* knock-down could be caused by aberrant splicing of glycolytic enzymes. Semi-quantitative RT–PCR analysis of selected glycolytic enzymes indeed revealed reduced splicing of *Gapdh2* and *PyK* mRNA, and lowered levels of both spliced and unspliced mRNAs for *Pyruvate dehydrogenase kinase* (*Pdk*) upon knock-down of *ecd* throughout the entire wing disc ($C765 > ecd^{RNAi}$, Fig EV2A–D). The expression and splicing of the translation initiation factor *eIF-4a* was not affected by loss of Ecd, however, indicating that there is not a general block in splicing efficiency (Appendix Fig S3E–E″). We observed the same effect on *Gapdh2* splicing upon knock-down of the interacting partner of Ecd, *brr2*, a core member of the splicing machinery ($C765 > brr2^{RNAi}$, Appendix Fig S4A and A′).

Lastly, we assessed the phenotypic consequences of Ecd loss of function on the adult wing morphology. The $ecd^{RNAi}$ is problematic for this purpose, given that long-term induction produces adults with vestigial wings (Claudius *et al*, 2014) (Fig 2D). Thus, we devised an alternative approach of over-expressing a dominant-negative allele (see Methods). Expression of this construct has the same effect on *Gapdh2* splicing as $ecd^{RNAi}$ ($C765 > ecd^{DN}$, Appendix Fig S4B and B′). In other respects, this construct produces milder phenotypes than $ecd^{RNAi}$, giving rise to flies with smaller wings that are proportionally broader along the A/P axis—similar to the knock-down of *Gapdh*, as well as other glycolytic enzymes (Fig 2A–C′, E and F, Appendix Fig S2).

Taken together, our results are consistent with the hypothesis that the glycolysis-promoting function of Ecd shown in yeast and in human cells is conserved in *Drosophila*. Ecd is required autonomously for tissue growth, correct splicing of several glycolytic enzymes, maintenance of steady-state levels of ATP and properly proportioned adult wings. While we cannot rule out the possibility that loss of Ecd also affects genes unrelated to glycolysis, the phenotypic similarity with respect to ATP levels and adult wing phenotype between Gapdh and Ecd loss of function suggests that they act functionally in the same pathway and that aberrant splicing of *Gapdh2* contributes to the growth defect in the wing.

## Glycolytic activity influences Smoothened membrane accumulation

The wing shape phenotype exhibited by Ecd and glycolytic enzymes, such as Gapdh, suggests that they can affect the signalling systems that drive tissue growth and patterning. We next explored the coupling between metabolism and Hh pathway activity by

**Figure 3. Glycolytic activity regulates Smoothened membrane accumulation.**

A–B′ Time-controlled knock-down of *ecd* in the dorsal compartment of the wing discs (see Fig 1C for the expression pattern of $apGal^{ts}$). IF of control (A, A′) and $apGal^{ts} > ecd^{RNAi}$ (B, B′) wing discs, stained for the Hedgehog (Hh) pathway components Smoothened (Smo, A, B) and $Ci_{155}$ (A′, B′). Next to the images are quantifications of the respective staining in the dorsal versus ventral compartments of control ($n = 5$) and $apGal^{ts} > ecd^{RNAi}$ ($n = 5$) wing discs. Graphs show mean (thick line) $\pm$ SD (thin lines). Dashed yellow lines indicate the position of the A/P boundary. Statistical analyses (*t*-test) revealed significant differences in Smo and $Ci_{155}$ expression in the dorsal compartment between control and $apGal^{ts} > ecd^{RNAi}$ (Smo: $P \leq 0.001$, $Ci_{155}$: $P \leq 0.001$) wing discs. Thus, loss of Ecd elevates Smo levels on the plasma membrane and increases the range of $Ci_{155}$ stabilization in the anterior compartment. Wing discs were analysed 48 h after RNAi induction. Scale bars = 50 μm.

C–E′ Time-controlled knock-down of metabolic enzymes in the dorsal compartment of the wing disc. IF of $apGal^{ts} > Pfk^{RNAi}$, $apGal^{ts} > Gapdh^{RNAi}$ and $apGal^{ts} > Glo1^{RNAi}$ wing discs, stained for Smo (C, D, E) and $Ci_{155}$ (C′, D′, E′). Next to the images are quantifications of the respective staining in the dorsal versus ventral compartments of $apGal^{ts} > Pfk^{RNAi}$ ($n = 5$), $apGal^{ts} > Gapdh^{RNAi}$ ($n = 5$) and $apGal^{ts} > Glo1^{RNAi}$ ($n = 6$) wing discs. Graphs show mean (thick line) $\pm$ SD (thin lines). Dashed yellow lines indicate the position of the A/P boundary. Statistical analyses (*t*-test) revealed significant differences in Smo and $Ci_{155}$ expression in the dorsal compartment between control and $apGal^{ts} > Pfk^{RNAi}$ (Smo: $P \leq 0.001$, $Ci_{155}$: $P \leq 0.01$), $apGal^{ts} > Gapdh^{RNAi}$ (Smo, $Ci_{155}$: $P \leq 0.05$) or $apGal^{ts} > Glo1^{RNAi}$ (Smo: $P \leq 0.001$, $Ci_{155}$: $P \leq 0.05$) wing discs. Thus, loss of Pfk, Gapdh or Glo1 elevates Smo levels on the plasma membrane and increases the range of $Ci_{155}$ stabilization in the anterior compartment. Wing discs were analysed 120 h after RNAi induction. Scale bars = 50 μm.

Source data are available online for this figure.

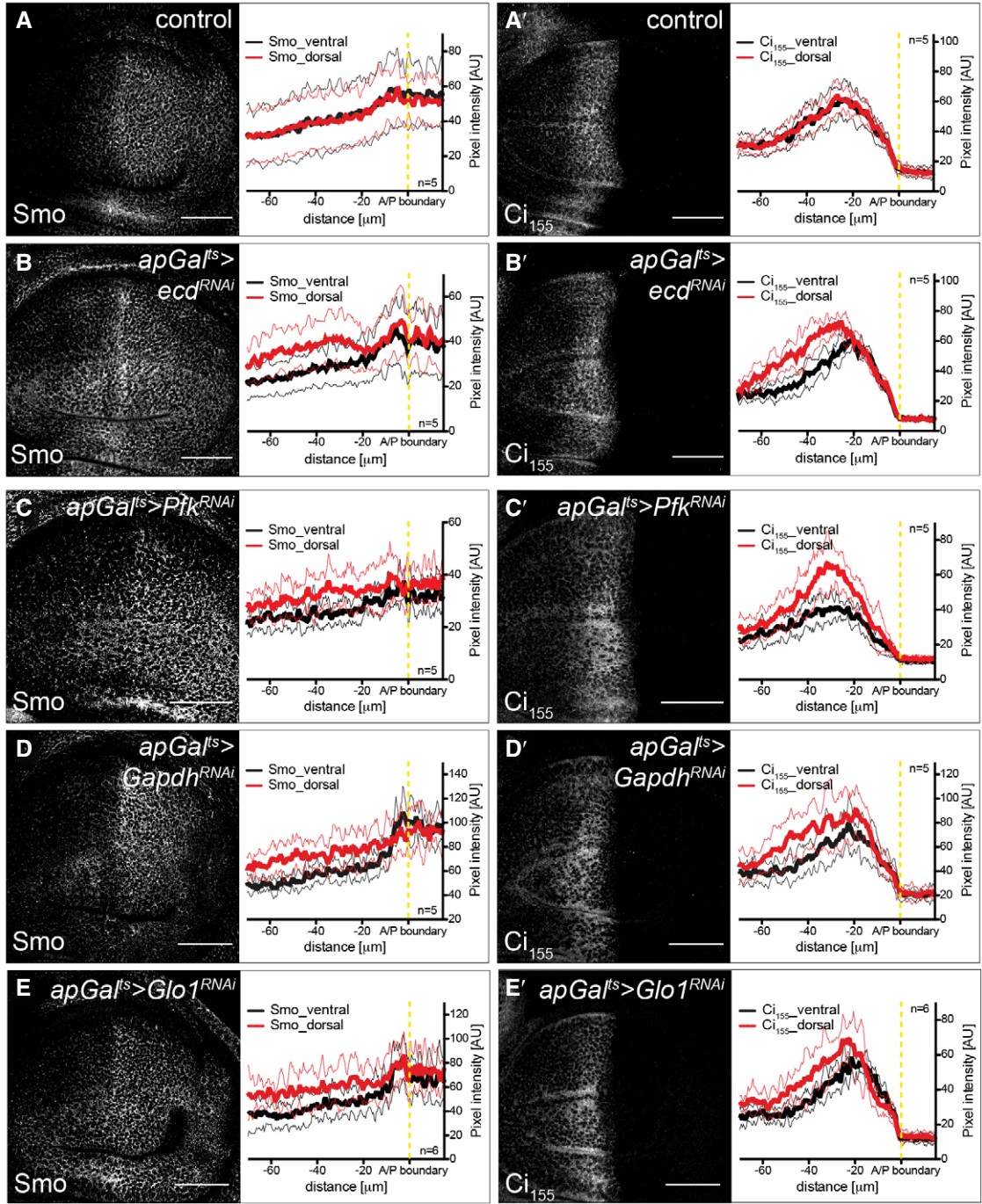

**Figure 3.**

determining how Smo membrane localization is affected by loss of glycolytic activity. We found that loss of Ecd autonomously elevates Smo accumulation on the basolateral membrane (Fig 3A and B) without altering levels or splicing of *smo* mRNA (Fig EV3). Additionally, loss of Ecd stabilizes $Ci_{155}$ in most of the anterior compartment, except at the most anterior end (Fig 3A′ and B′). Expression of the dominant-negative Ecd construct ($apGal^{ts} > ecd^{DN}$, Fig EV4A–B″) had the same effect. Also, we found that inducing RNAi against the splicing component *brr2* similarly affected Smo and $Ci_{155}$

($apGal^{ts} > brr2^{RNAi}$, Fig EV4C–C″), indicating that the function of Ecd as a splicing regulator is likely important for its role in affecting Hh signalling. To confirm that glycolysis specifically could cause such effect on Hh signalling, we again used the knock-downs of single metabolic enzymes. Indeed, we found that RNAi directed against *Pfk*, *Gapdh* and *Glo1* also induced Smo accumulation and $Ci_{155}$ stabilization (Fig 3C–E′). Taken together, these data suggest that interfering with glycolysis increases the stability of Smo and its ability to block the processing of $Ci_{155}$ to its repressor form.

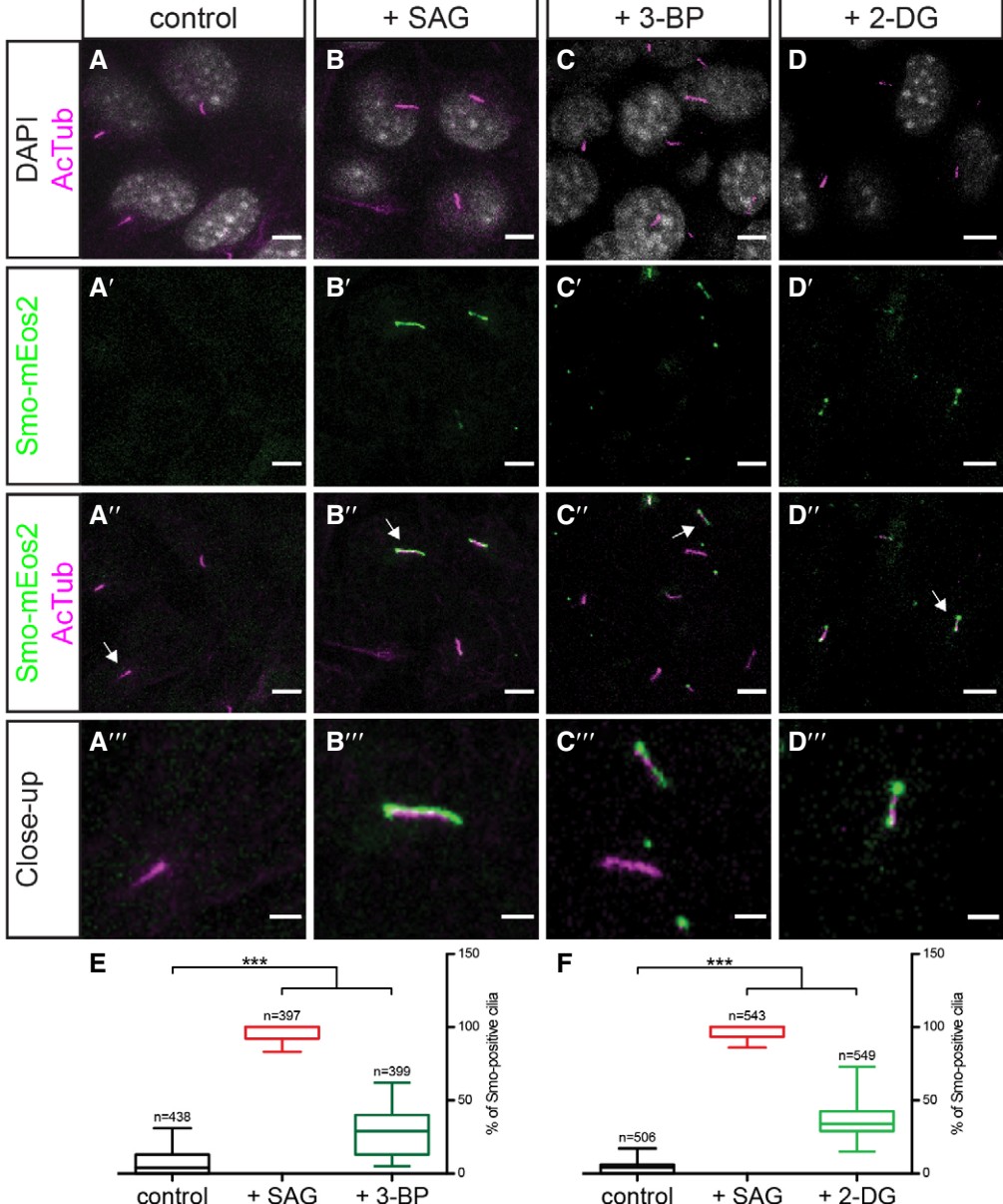

**Figure 4. Glycolytic activity regulates Smo localization to the primary cilium in mammalian cells.**

A–D‴ IF of NIH3T3 cells stably expressing Smo-mEos2 treated with mock (control, A-A‴), the Smo agonist (+ SAG, B-B‴) or the glycolytic inhibitors 3-bromopyruvate (+3-BP, C-C‴) and 2-deoxyglucose (+ 2-DG, D-D‴). DAPI staining, shown in grey, was used to visualize cell nuclei. Acetylated tubulin (AcTub), used as a ciliary marker, is shown in magenta; Smo-mEos2 is shown in green. Zoomed-in images of cilia marked by white arrows in A″, B″, C″ and D″ are shown in A‴, B‴, C‴ and D‴. Scale bars = 5 μm, but 2 μm in A‴, B‴, C‴ and D‴.

E, F Box plot showing quantification of Smo-mEos2 ciliary localization in NIH3T3 cells upon 3-BP (E) and 2-DG (F) treatment, respectively. Lower and upper hinges correspond to the first and third quartiles; vertical lines extend to the minimum and maximum values. Sample size is indicated for each perturbation. Treatment with glycolytic inhibitors induces Smo-mEos2 localization to the primary cilium. *t*-test, \*\*\**P* ≤ 0.001.

Source data are available online for this figure.

We next investigated whether the influence of glycolysis on Smo membrane localization is conserved across phyla. In mammalian cells, activation of Hh signalling causes Smo to accumulate in the primary cilium (Corbit *et al*, 2005; Rohatgi *et al*, 2007), rather than on the basolateral membrane as in *Drosophila*. We inhibited glycolysis pharmacologically by treating NIH3T3 cells expressing Eos-tagged Smo (Smo-mEos2) (Kim *et al*, 2014) with either 3-

bromopyruvate (3-BP) or 2-deoxyglucose (2-DG). 2-DG inhibits hexokinase, while 3-BP inhibits both hexokinase and Gapdh (Gana-pathy-Kanniappan *et al*, 2013). In cells treated with 2-DG or 3-BP, Smo-mEos2 accumulates in the primary cilium despite the fact that Hh ligands are not present (Fig 4A–F), indicating that glycolytic metabolism is required to keep Smo out of the primary cilium in mammalian cells in the absence of Hh ligands. This result indicates

that the effect of cell metabolism on Smo accumulation is conserved across phyla.

## Loss of Ecdysoneless sensitizes cells to Hedgehog

To examine whether the effects of metabolism on *Drosophila* Smo accumulation does not require the Hh ligand, we first investigated how the loss of Ecd influences the distribution of Hh. Immunostaining revealed no obvious differences in Hh protein levels or distribution upon *ecd* knock-down in the dorsal compartment (Fig 5A and

B). Thus, loss of Ecd does not stabilize Smo protein by increasing the amount of Hh released into the anterior compartment. We then specifically knocked down *ecd* in the posterior Hh-producing cells and observed no effect on Smo levels in the anterior compartment (Appendix Fig S5), again suggesting that Ecd does not act by regulating Hh production. In contrast, anterior knock-down of *ecd* stabilized both Smo and $Ci_{155}$ (Appendix Fig S6), suggesting that Ecd normally acts directly in receiving tissue to prevent ectopic Smo activation. These results are consistent with the early observation that whole animal mutants of *ecd* can be rescued by driving Ecd

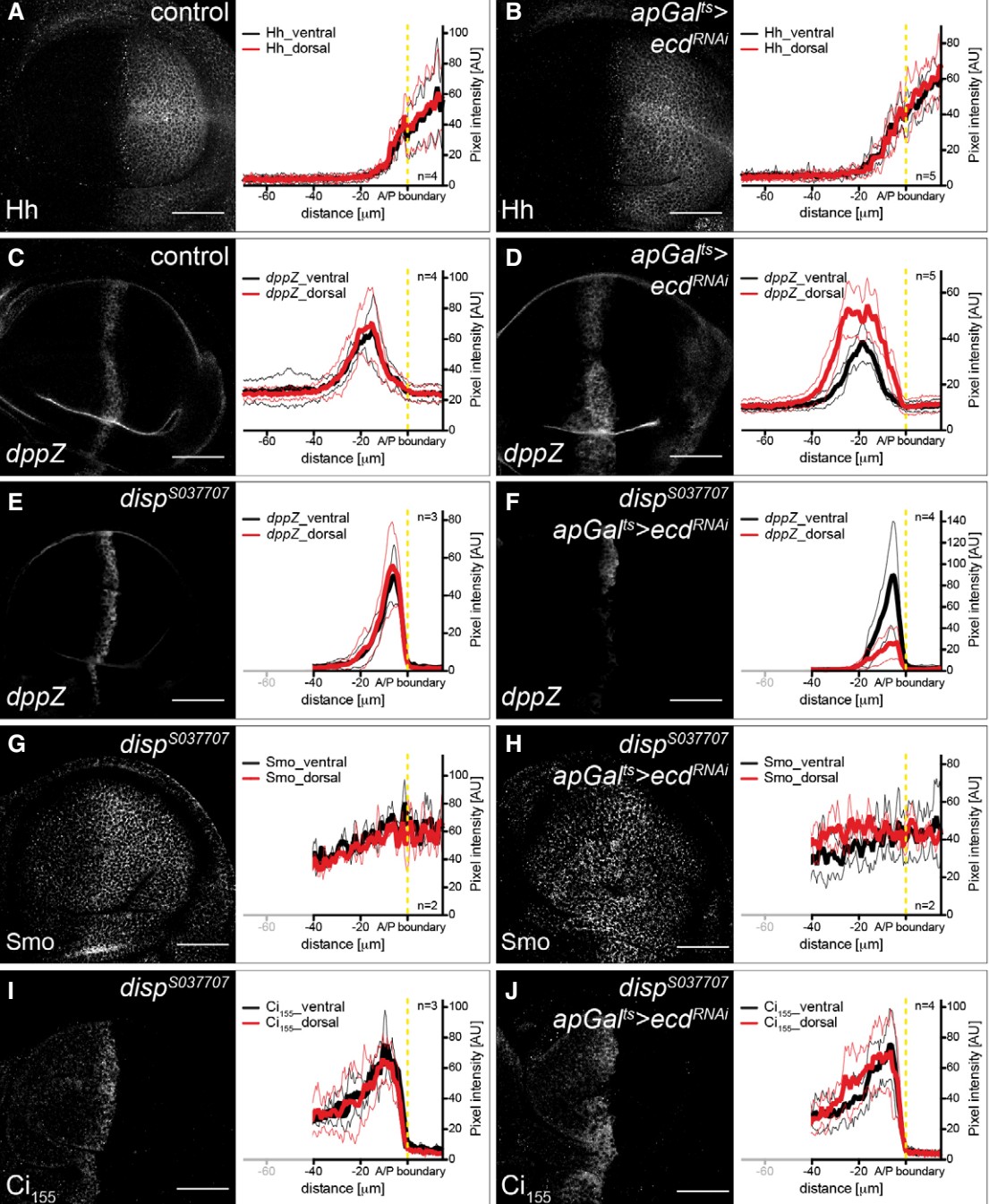

**Figure 8.**

**Figure 5.  Loss of Ecd sensitizes tissue to Hh by stabilizing Smo.**

A, B   Time-controlled knock-down of *ecd* in the dorsal compartment of the wing discs (see Fig 1C for the expression pattern of *apGal^ts*). IF of control (A) and *apGal^ts*>*ecd^RNAi* (B) wing discs, stained for Hh. Next to the images are quantifications of the respective staining in the dorsal versus ventral compartments of control (n = 4) and *apGal^ts*>*ecd^RNAi* (n = 5) wing discs. Graphs show mean (thick line) ± SD (thin lines). Dashed yellow lines indicate the position of the A/P boundary. Statistical analysis (*t*-test) revealed no significant differences in Hh expression in the dorsal compartment between control and *apGal^ts*>*ecd^RNAi* wing discs. Thus, loss of Ecd does not affect Hh production and spreading. Wing discs were analysed 48 h after RNAi induction. Scale bars = 50 μm.

C–F   Time-controlled knock-down of *ecd* in the dorsal compartment of *dispatched* mutant wing discs (*disp^S037707*, *apGal^ts*>*ecd^RNAi*, see Fig 1C for the expression pattern of *apGal^ts*). IF of control (C), *apGal^ts*>*ecd^RNAi* (D), *disp^S037707* (E) and *disp^S037707*, *apGal^ts*>*ecd^RNAi* (F) wing discs expressing the *decapentaplegic* (*dpp*) reporter gene *dpp-lacZ* (*dppZ*), stained for *dppZ*. Next to the images are quantifications of the respective stainings (n = 4 (C, F), n = 5 (D), n = 3 (E)). Graphs show mean (thick line) ± SD (thin lines). Dashed yellow lines indicate the position of the A/P boundary. Statistical analyses (*t*-test) revealed a significant difference in *dppZ* expression in the dorsal compartment between control and *apGal^ts*>*ecd^RNAi* (P ≤ 0.01) (C versus D), and *disp^S037707* and *disp^S037707*, *apGal^ts*>*ecd^RNAi* (P ≤ 0.01) wing discs (E versus F). Loss of Ecd elevates the expression of *dppZ* in a broader than normal stripe. However, lack of Ecd function alone is not sufficient to trigger *dpp* expression, which instead depends on the presence of Hh. Wing discs were analysed 48 h after RNAi induction. Scale bars = 50 μm.

G–J   IF of *disp^S037707* (G, I) and *disp^S037707*, *apGal^ts*>*ecd^RNAi* (H, J) wing discs, stained for Smo (G, H) and $Ci_{155}$ (I, J). Next to the images are quantifications of the respective stainings (n = 2 (G, H), n = 3 (I), n = 4 (J)). Graphs show mean (thick line) ± SD (thin lines). Dashed yellow lines indicate the position of the A/P boundary. Although statistical analyses (*t*-test) revealed no significant differences in Smo and $Ci_{155}$ expression in the dorsal compartment between control and *disp^S037707*, *apGal^ts*>*ecd^RNAi* wing discs, loss of Ecd slightly induces an increased Smo accumulation and $Ci_{155}$ stabilization independently of Hh. Wing discs were analysed 48 h after RNAi induction. Scale bars = 50 μm.

Source data are available online for this figure.

expression specifically in Hh-receiving but not producing cells (Gaziova *et al*, 2004).

To test whether Hh ligand contributes to the effect of Ecd on the Hh signalling pathway, we examined target gene expression in a *dispatched* (*disp*) mutant background. Disp is required for the release of Hh ligands from producing cells (Burke *et al*, 1999). In the presence of Disp, loss of Ecd broadens the range of *dppZ* expression (Fig 5C and D). However, in the absence of Disp, when Hh ligands cannot be released, *ecd* knock-down actually reduces *dppZ* expression (Fig 5E and F). This result indicates that Hh ligand is required for target gene activation upon *ecd* knock-down. Even though target gene activation is reduced, however, knock-down of *ecd* in a *disp* mutant does not reduce Smo membrane accumulation or $Ci_{155}$ stabilization but actually slightly increases it (Fig 5G–J). In this case, how could the knock-down of *ecd* affect target gene activation but not Ci and Smo? Our previous published results suggest that stabilization and activation of $Ci_{155}$ are differently affected by lipoprotein-associated (Lpp-associated) Hh and sterol-free Hh (HhN) (Khaliullina *et al*, 2009; Palm *et al*, 2013). In this way, the phenotype caused by loss of Ecd resembles that of flooding wing discs with Lpp-associated Hh (and the phenotype caused by *lpp* knock-down)—it can stabilize $Ci_{155}$ but must cooperate with another form of Hh to activate $Ci_{155}$ for target gene activation. When imaginal discs are exposed to Lpp-associated Hh, they require much lower levels of HhN to fully activate target genes (Palm *et al*, 2013). Similarly, stabilization of $Ci_{155}$ by *ecd^RNAi* appears to sensitize imaginal cells to the endogenous Hh emanating from the posterior compartment—allowing the same levels of Hh to activate Ci-dependent gene expression over a broader range.

In total, our data from *Drosophila* suggest that, as in mammalian cells, perturbation of glycolysis affects the Hh signalling pathway without requiring the Hh ligand. We note, however, that the effects of loss of Ecd in *Drosophila* are more pronounced in the presence of ligand, indicating that Hh activation of the pathway contributes to the Ecd phenotype.

**Plasma membrane potential depends on glycolysis**

How could changes in metabolism stabilize Smo independently of the Hh ligand? Several small lipidic molecules are known to

modulate Smo activity (Bijlsma *et al*, 2006; Myers *et al*, 2013, 2017; Khaliullina *et al*, 2015; Huang *et al*, 2016; Byrne *et al*, 2018). Many small metabolites depend on $Na^+$-driven transporters for their uptake into cells (Tsukaguchi *et al*, 1999; Ritzel *et al*, 2001; Wright *et al*, 2004; Gurav *et al*, 2015). Indeed, Ptc activity has recently been shown to be $Na^+$-dependent (Myers *et al*, 2017). The transmembrane $Na^+$ gradient that drives these transport processes is generated by the $Na^+/K^+$-ATPase. This electrogenic pump produces opposing gradients of $Na^+$ and $K^+$ across the plasma membrane and establishes the transmembrane potential. Experiments in other cells suggest that the activity of the $Na^+/K^+$-ATPase is coupled to glycolysis. Two sequential glycolytic enzymes, Gapdh and Pgk, are present in complexes with the $Na^+/K^+$-ATPase and may locally supply it with ATP (Mercer & Dunham, 1981; Balaban & Bader, 1984; Lynch & Balaban, 1987; James *et al*, 1999). We therefore wondered whether glycolysis might influence Hh signalling through its effects on the plasma membrane potential. To investigate this idea, we asked how loss of Ecd, Pfk or Gapdh affected staining with the plasma membrane potential-sensitive dye $DiBAC_4(3)$. Indeed, induction of RNAi against either *ecd* or *Gapdh* in the dorsal compartment of the wing disc lowers plasma membrane potential, compared to the control ventral compartment (Fig 6A–A‴), as does *Pfk^RNAi* (Fig EV5). This result suggests that glycolysis may increase $Na^+/K^+$-ATPase activity in the wing disc.

**Cellular N-acylethanolamide uptake depends on glycolysis and cation gradients**

N-acylethanolamides delivered by the *Drosophila* lipoprotein, Lipophorin (Lpp), are required for Ptc-mediated destabilization of Smo in the wing disc (Khaliullina *et al*, 2009, 2015). Since the Hh pathway is affected similarly by reducing glycolysis and by loss of Lpp, we tested whether the depolarization of the plasma membrane caused by *ecd* or *Gapdh* knock-down might interfere with cellular uptake of N-acylethanolamides. To study N-acylethanolamide trafficking, we synthesized a photoactivatable and clickable N-acylethanolamide analogue (PAC-NAE). PAC-NAE contains a C15 fatty acid featuring diazirine and alkyne modifications (Fig 6B). After incubation with tissue explants, PAC-NAE intracellular localization can be

visualized by attaching azide-modified fluorescent dyes using click chemistry (Haberkant *et al*, 2013; Gaebler *et al*, 2016; Höglinger *et al*, 2017; Müller *et al*, 2020). To study N-acylethanolamide trafficking, we loaded PAC-NAE onto preparations of *Drosophila* Lpp at levels comparable to those observed in circulation and applied these lipoproteins to explanted wing discs. We then visualized the tissue distribution of PAC-NAE after attachment of Alexa 555 azide. PAC-NAE accumulation is autonomously reduced by knock-down of either *ecd* or *Gapdh* in the dorsal compartment (Fig 6C–C‴). In

*Gapdh^RNAi* tissue, the reduction of PAC-NAE uptake is less pronounced near the A/P boundary (Fig 6C‴). Overall, these data indicate that reducing glycolysis inhibits uptake of PAC-NAE. To determine whether plasma membrane depolarization would be sufficient to account for this inhibition, we applied PAC-NAE to wing discs that were incubated with gramicidin A (gA), which forms cation-permeable channels that dissipate the plasma membrane potential (Fig 6D–D″). GA treatment strongly reduces uptake of PAC-NAE compared to control discs (Fig 6E–E″). Taken together,

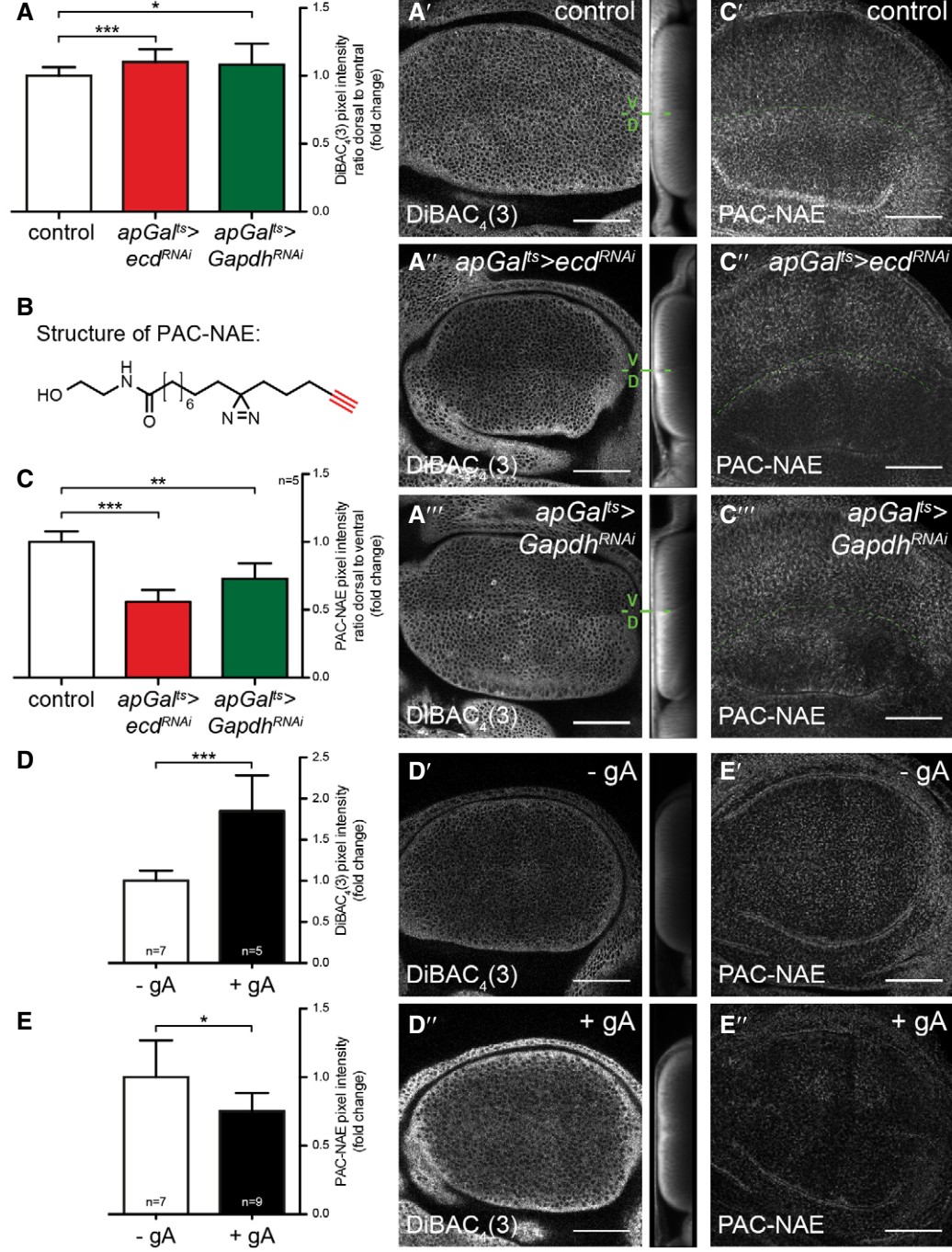

**Figure 6.**

◀ **Figure 6.  Loss of Ecd or Gapdh depolarizes the plasma membrane and reduces the uptake of the N-acylethanolamide analogue PAC-NAE.**

A–A‴   Time-controlled knock-down of *ecd* or *Gapdh* in the dorsal compartment of the wing discs (see Fig 1C for the expression pattern of *apGal$^{ts}$*). The plasma membrane potential was detected using DiBAC$_4$(3). The fluorescence intensity of DiBAC$_4$(3) increases upon plasma membrane depolarization. (A) The change in plasma membrane potential upon genetic perturbation was quantified as the fold change in the ratio of DiBAC$_4$(3) signal intensity in the dorsal to ventral compartment (control (n = 19), *apGal$^{ts}$>ecd$^{RNAi}$* (n = 36) and *apGal$^{ts}$>Gapdh$^{RNAi}$* (n = 18) wing discs). Error bars indicate ± SD. *t*-test, *P ≤ 0.05, ***P ≤ 0.001. (A′–A″) Representative images of DiBAC$_4$(3) assay in control (A′), *apGal$^{ts}$>ecd$^{RNAi}$* (A″) and *apGal$^{ts}$>Gapdh$^{RNAi}$* (A‴) wing discs 48 h or 120 h after RNAi induction, respectively. V/D indicates the boundary between the ventral and dorsal compartments. Next to the maximal projections are shown the sum projections of cross-sections along the A/P axis. Loss of Ecd or Gapdh alters the plasma membrane potential. Scale bars = 50 μm.

B   Structure of the N-acylethanolamide analogue PAC-NAE. PAC-NAE contains a C15 fatty acid featuring diazirine and alkyne (red) modifications.

C–C‴   Time-controlled knock-down of *ecd* or *Gapdh* in the dorsal compartment of the wing discs (see Fig 1C for the expression pattern of *apGal$^{ts}$*). PAC-NAE uptake was quantified as the ratio of PAC-NAE signal intensity in the dorsal to ventral compartments. Shown is the fold change from control in *apGal$^{ts}$>ecd$^{RNAi}$* and *apGal$^{ts}$>Gapdh$^{RNAi}$* wing discs (C). For each condition, dorsal/ventral PAC-NAE signal intensity was quantified for 5 wing discs. Error bars indicate ± SD. *t*-test, **P ≤ 0.01, ***P ≤ 0.001. (C′-C″) Representative images of PAC-NAE uptake in control (C′), *apGal$^{ts}$>ecd$^{RNAi}$* (C″) and *apGal$^{ts}$>Gapdh$^{RNAi}$* (C‴) wing discs at 48 h or 120 h after RNAi induction, respectively. Loss of Ecd or Gapdh blocks the uptake of PAC-NAE. Dashed green lines mark the D/V boundary. Scale bars = 50 μm.

D–E″   Dissipation of the plasma membrane potential upon treatment of wing discs with gramicidin A (gA). (D) Quantification of DiBAC$_4$(3) signal intensity in mock-treated (− gA, n = 7) and gA-treated (+ gA, n = 5) wing discs. (D′, D″) Representative images of DiBAC$_4$(3) assay in mock-treated (D′) and gA-treated (D″) wing discs. Next to the maximal projections are shown sum projections of cross-sections along the A/P axis. (E) Quantification of PAC-NAE signal intensity in mock-treated (n = 7) and gA-treated (n = 9) wing discs, shown as fold change from control (mock-treated). (E′-E″) Representative images of PAC-NAE uptake in mock-treated (E′) and gA-treated (E″) wing discs. GA treatment strongly reduces the uptake of PAC-NAE. Error bars indicate ± SD. *t*-test, *P ≤ 0.05, ***P ≤ 0.001. Scale bars = 50 μm.

Source data are available online for this figure.

these data suggest that reducing glycolysis stabilizes Smo and Ci$_{155}$ in a Hh-independent fashion by altering the plasma membrane potential and inhibiting N-acylethanolamide uptake.

## Discussion

Patterns of growth and differentiation during development are functionally associated with changes in cell metabolism (Kaelin & McKnight, 2013; Shyh-Chang *et al*, 2013; Buck *et al*, 2015; Oginuma *et al*, 2017; Müller *et al*, 2020). Shifts between oxidative and glycolytic metabolism can alter the balance of differentiated T-cell and macrophage populations (Buck *et al*, 2015), and regulate the balance of proliferation and differentiation in the mouse cerebellum (Gershon *et al*, 2013) and in *Drosophila* neuroblasts (Homem *et al*, 2014). While it is clear that metabolite levels can exert strong effects on histone acetylation and methylation (Agathocleous & Harris, 2013; Kaelin & McKnight, 2013; Shyh-Chang *et al*, 2013), many mysteries remain about how metabolic shifts cause changes in gene expression programs.

Many intriguing studies have also linked tissue growth, patterning and regeneration to changes in plasma membrane potential (Levin, 2007). For example, frog and chick embryos develop differences in transmembrane potential on the left and right sides of the primitive streak that are important for left–right symmetry breaking (Levin, 2007). In planaria, plasma membrane potential is lower in the head than in the tail, and reducing plasma membrane potential during regeneration leads to two-headed worms (Beane *et al*, 2011). It has been challenging to identify the molecular mechanisms underlying such dramatic developmental changes.

Here, we establish a mechanistic link between cell metabolism, plasma membrane potential and developmental signalling. We show that changes in cell metabolism alter the plasma membrane potential and that altered plasma membrane potential directly influences the activity of the Hh pathway. Interestingly, other studies have

shown that inhibiting glycolysis phenocopies loss of Wnt signalling during vertebrate somitogenesis (Oginuma *et al*, 2017) and alters Notch signalling in the *Drosophila* wing disc (Saj *et al*, 2010), although the mechanistic bases of these effects are not known. Thus, it appears that the metabolic state of developing tissues may broadly influence how they respond to many morphogenetic signals. It will be interesting to investigate whether changes in plasma membrane potential might affect the activity of these pathways.

Our findings highlight an important role for glycolysis in maintaining plasma membrane potential in the wing disc. The activity of the $Na^+/K^+$-ATPase is key to generating plasma membrane potential—it consumes ATP to pump 3 $Na^+$ out of the cell for every 2 $K^+$ that enter the cell (Skou & Esmann, 1992). Resting plasma membrane potential is further tuned by channels that control cellular permeability to specific anions and cations (Ashmore & Meech, 1986). Previous studies in cultured cells have suggested that the activity of the $Na^+/K^+$-ATPase is coupled to glycolysis. Glycolysis appears to be more efficient than oxidative phosphorylation in supplying ATP to the $Na^+/K^+$-ATPase (Balaban & Bader, 1984; James *et al*, 1999), and work in erythrocytes suggests this could be due to its physical association with a membrane pool of Gapdh and Pgk (Mercer & Dunham, 1981). Our finding that *ecd*, *Gapdh* and *Pfk* knock-down each alters the plasma membrane potential in the wing disc suggests that $Na^+/K^+$-ATPase activity in this tissue may also be preferentially supported by glycolysis. It will be interesting to explore whether a subset of glycolytic enzymes associates with the *Drosophila* $Na^+/K^+$-ATPase.

We show that the plasma membrane potential is necessary for the cellular uptake of N-acylethanolamide. Endocannabinoid family lipids delivered to the wing disc by circulating lipoproteins are required to prevent Smo activation in the absence of Hh ligands (Khaliullina *et al*, 2009, 2015; Palm *et al*, 2013). The potential energy available from the transmembrane $Na^+$ gradient generated by the $Na^+/K^+$-ATPase is exploited by many transmembrane transporters to move a wide variety of nutrients and other small molecules into cells (Tsukaguchi *et al*, 1999; Ritzel

*et al*, 2001; Wright *et al*, 2004; Bergeron *et al*, 2013), and our data are consistent with the idea that N-acylethanolamide uptake depends on such a $Na^+$-coupled transporter. Interestingly, recent work suggests that the repressive activity of Ptc relies on the $Na^+$ gradient (Myers *et al*, 2017). Several studies suggest a key role for cholesterol in Smo activation and suggest that Ptc might deplete cholesterol from the vicinity of Smo (Myers *et al*, 2013, 2017; Huang *et al*, 2016; Byrne *et al*, 2018; Zhang *et al*, 2018). It has also been proposed that Ptc regulates trafficking of an inhibitor that competes with cholesterol for Smo binding (Myers *et al*, 2017). It will be interesting to explore whether Ptc affects N-acylethanolamide uptake, or whether these lipids depend on other membrane potential-dependent transporters.

In summary, we have revealed a molecular mechanism through which glycolysis affects Hh pathway activity in both *Drosophila* and mammalian cells. Changes in glycolysis leading to lower ATP levels depolarize the plasma membrane and thereby reduce uptake of Hh pathway inhibitory lipids. This work is an important step towards understanding how membrane potential influences development and regeneration. Furthermore, this work provides mechanistic insight into how cellular metabolism can be connected to the conserved developmental signalling pathways that regulate tissue growth and patterning.

# Materials and Methods

### *Drosophila* stocks and genetics

The following stocks were used: wild-type Oregon-R-C (BDSC, #5), *disp*[S037707] (Burke *et al*, 1999), *dpp-lacZ*[BS3.0] (Blackman *et al*, 1991), *fosEcd* (Sarov *et al*, 2016), *Gapdh1::GFP*[V5] (this study), *GFP*[Myc]::*Gapdh2* (this study), *ubi-AT1.03NL* (this study), *ap-Gal4* (BDSC, #3041), *C765-Gal4* (BDSC, #36523), *dpp-Gal4* (BDSC, #1553), *en (105)-Gal4* (Eugster *et al*, 2007), *tub-Gal80*[ts] (BDSC, #7019 or #7017), *UAS-Ald*[RNAi] (VDRC, #101339, #27541 or #47668), *UAS-brr2*[RNAi] (VDRC, #110666), *UAS-ecd*[RNAi] (BDSC, #41676 or VDRC, #103145), *UAS-Eno*[RNAi] (VDRC, #110090), *UAS-Gapdh*[RNAi] (BDSC, #26302), *UAS-Gapdh1*[RNAi] (BDSC, #36842, #62212; VDRC #31631 or #100596), *UAS-Gapdh2*[RNAi] (VDRC, #106562, #23645, or #23646), *UAS-Glo1*[RNAi] (VDRC, #26832), *UAS-Hex-A*[RNAi] (BDSC, #35155), *UAS-Pfk*[RNAi] (VDRC, #3017), *UAS-Pgk*[RNAi] (BDSC, #33633; VDRC, #110081 or #33798), *UAS-PyK*[RNAi] (BDSC, #35218 or VDRC, #49533), *UAS-Treh*[RNAi] (BDSC, #50585, #51810 or VDRC, #30731), *UAS-ecd::TAG* (this study) and *UAS-ecdQ650*::*TAG* (this study).

For all experiments, flies were raised on a standard food containing cornmeal, malt, sugar beet syrup, yeast and soy meal. If not specified otherwise, experiments were performed at 25°C. For all experiments using *ap-Gal4*, *tub-Gal80*[ts] (*apGal*[ts]), larvae were grown at 20°C until day 1–6 after an overnight egg collection (AEL), then transferred to 30°C and dissected 30, 48, 96 or 120 h later. For the experiment using *dpp-Gal4*, *tub-Gal80*[ts] (*dppGal*[ts]), larvae were grown at 20°C until day 5 AEL, then transferred to 30°C and dissected 48 h later. For all experiments using *C765-Gal4*, larvae were transferred to 29°C immediately AEL at 25°C. In all cases, respective controls were handled in the same manner.

The genotypes of the wing discs and adult wings analysed in this paper are described in Appendix Table S1.

### Generation of transgenic flies

Sequence information of the oligonucleotides used for the generation of transgenic flies is listed in Appendix Table S2.

The FRET-based ATP sensor *ubi-AT1.03NL* line was generated as follows. The ATP sensor sequence was amplified from the pUAST-AT1.03NL plasmid (a gift from T. Uemura, Graduate School of Biostudies, Kyoto University, Kyoto, Japan) (Tsuyama *et al*, 2013) using the primers AT_NL_AvrII_f and AT_NL_SpeI_r, and cloned into the pCM43-ubi-SV40 vector (Aigouy *et al*, 2010). Transgenic flies were generated by phiC31-mediated integration into the VK00033 landing site by BestGene (Chino Hills, CA, USA).

The CRISPR/Cas9 *Gapdh1::GFP* and *GFP::Gapdh2* lines were generated according to Port *et al* (2014). Briefly, the optimal protospacer-adjacent motif (PAM) was selected with the online tool (http://targetfinder.flycrispr.neuro.brown.edu/) (Gratz *et al*, 2014) and located 11 bp downstream of the start codon of *Gapdh1* and *Gapdh2*, respectively. For *Gapdh1::GFP*, oligos for gRNAs (Gapdh1_sense and Gapdh1_antisense) were ordered from Sigma-Aldrich (Munich, Germany) and cloned into pCFD3 vector (Port *et al*, 2014). Two homology arms (~1.2 kb each) were amplified from gDNA generated from wild-type flies by PCR using the primer pairs Gapdh1_left_f & Gapdh1_left_r and Gapdh1_right_f & Gapdh1_right_r, and the sGFP-tag was amplified from the fosEcd construct encoding C-terminally tagged Ecd (Sarov *et al*, 2016) using the primers Gapdh1_sGFP_f & Gapdh1_sGFP_r. The primers Gapdh1_sGFP_r and Gapdh1_right_f additionally contained a V5-tag sequence. To generate the construct for homology-directed repair (HDR), the amplified PCR products encoding the left homology arm, sGFP- and V5-tag, and the right homology arm were cloned together into pBluescript-KS(+) (a gift from E. Knust, MPI-CBG, Dresden, Germany) by Gibson assembly. For *GFP::Gapdh2*, the following oligos for gRNAs (Gapdh2_sense and Gapdh2_antisense) were ordered and cloned into pCFD3 vector. Two homology arms (~1.1 kb each) were amplified from gDNA generated from wild-type flies by PCR using the primer pairs NotI_Gapdh2_f & NcoI_Gapdh2_r and EarI_Gapdh2_f & KpnI_Gapdh2_r, and the sGFP-tag was amplified from the fosEcd construct encoding C-terminally tagged Ecd (Sarov *et al*, 2016) using the primers NcoI_Gapdh2_sGFP_f & EarI_Gapdh2_sGFP_r. The primers EarI_Gapdh2_sGFP_r and EarI_Gapdh2_f additionally contained a 3xMyc-tag sequence. To generate the construct for homology-directed repair (HDR), the amplified PCR products encoding the left homology arm, sGFP- and 3xMyc-tag, and the right homology arm were cloned together into pBluescript-KS(+) by standard molecular biology techniques. For both *Gapdh1::GFP* and *GFP::Gapdh2*, respective plasmids (gRNAs and HDR) were validated by sequencing and then injected into embryos expressing Cas9 under control of *nanos* promoter (BDSC, #54591). F0 flies were crossed to w[1118] line (BDSC, #3605), and F1 progeny was screened for GFP signal. The founder flies were verified by sequencing, and the genetic background was cleaned by outcrossing to w[1118] for three generations.

The *UAS-ecd* and *UAS-ecd*[DN] lines were generated as follows. For generation of the UAS-ecd::TAG (UAS-ecd) construct, the *ecd* gene was amplified from cDNA generated from wild-type flies using the primers NotI_cEcd_f and SacI_cEcd_r, and the protein tag (TAG = 2xTY1-SGFP-V5-preTEV-BLRP-3xFLAG) was amplified from the fosEcd using the primers SacI_fosEcd_f and KpnI_fosEcd_r. Both

fragments were cloned into the pUASTattB vector (Bischof *et al*, 2007) using standard ligation. The UAS-ecdQ650*::TAG (UAS-ecd$^{DN}$) construct is based on a lethal *ecd* allele, *ecd$^{l(3)23}$*, whose predicted mutant protein product is missing the C-terminal 35 amino acids (Gaziova *et al*, 2004). The *ecd* gene was amplified from cDNA generated from wild-type flies using the primers NotI_cEcd_f and Q650*_r, and the protein tag (TAG = 2xTY1-SGFP-V5-preTEV-BLRP-3xFLAG) was amplified from the fosEcd construct using the primers Q650*_f and KpnI_fosEcd_r. Both fragments were fused together using overlap extension PCR and then cloned into the pUASTattB vector using standard ligation. Before injections, all constructs were confirmed by sequencing. Transgenic flies were generated by phiC31-mediated integration into the VK00033 landing site (BDSC, #24871).

## Mammalian cell culture

NIH3T3/Smo-mEos2 cells (Kim *et al*, 2014) (a gift from P. Beachy, SUSM, Stanford, CA, USA) were grown to confluence on coverslips in DMEM (Thermo Fisher Scientific, #31966021) supplemented with 10% FBS (Thermo Fisher Scientific, #10270106), 1% penicillin/streptomycin (Thermo Fisher Scientific, #15140122) and 1% MEM non-essential amino acids (Thermo Fisher Scientific, #11140050) at 37°C. For SAG, 3-BP and 2-DG treatment, cells were then shifted to serum-deprived medium containing 0.5% FBS instead of 10% FBS and incubated for 24 h with 100 nM SAG (Merck, #566660), 25 mM of 2-DG (ROTH, #CN96.3) or 15 μM 3-BP (Sigma-Aldrich, #16490-10G), respectively, or without any compound at 37°C.

## Anti-Ecd antibody

For anti-Ecd, a peptide corresponding to amino acids 670–684 of *Drosophila* Ecd (FlyBase: CG5714-PA) was conjugated to keyhole limpet hemocyanin (KLH) and used to immunize guinea pigs (Eurogentec, Seraing, Belgium).

## Measurement of ATP using FRET-based ATP sensor

Wing discs from up-crawling third-instar larvae were dissected within 10 min in Grace's medium supplemented with 5% FBS and 20 nM 20-hydroxyecdysone (Dye *et al*, 2017). Dissected wing discs were mounted with their basal side up on glass-bottom dishes (MatTek Corporation, #P35G-1.0-20-C) with a double-sided tape spacer and immobilized with a Whatman™ Cyclopore™ track-etched polycarbonate membrane filter (GE Healthcare Life Sciences, #7062-2513). Imaging of *ubi-AT1.03NL* wing discs was performed on an Olympus IX81 microscope equipped with Yokogawa spinning disk (CSU-W1, Yokogawa) and Ixon Ultra EMCCD camera (Andor). The sensitized emission method was used to measure FRET. Wing discs were excited with a 445 nm laser twice in a sequential manner. Upon first excitation, emission of mse-CFP was collected using an HQ 480/40 bandpass filter. Upon second excitation, emission of cpVenus-FRET was collected using an HQ 542/27 filter. The emission of cpVenus-FRET has contributions from the energy transfer from mse-CFP and bleed-through of mse-CFP into the HQ 542/27 filter. The bleed-through was estimated by exciting wing discs expressing CFP-tagged human cytoplasmic β-actin (BDSC, #7064) under the control of *ubi-Gal4* and acquiring images through an HQ 480/40 filter ($I_D$) and its bleed-through in an

HQ 542/27 filter ($I_{bth}$). The fraction of FRET intensity contributed by bleed-through is given by,

$$\beta = \frac{I_{bth}}{I_D}$$

For the acquisition of a *z*-stack of a wing disc, donor and FRET images were sequentially acquired for each z-plane before moving the stage to the next plane. Wing discs were imaged using 30 z-planes 0.5 μm apart.

A custom-written MATLAB (MathWorks) script was used to estimate the FRET efficiency from the fluorescence images. Both donor and FRET images were smoothened using a $5 \times 5$ averaging kernel. The background was estimated from a region of the image without the wing disc. Donor ($I_D$) and FRET ($I_F$) images were background subtracted. Then, the FRET intensity was corrected for bleed-through as,

$$I_{FRET} = I_F - \beta I_D$$

The FRET efficiency (η) was calculated as,

$$\eta = \frac{I_{FRET}}{I_D + I_{FRET}}$$

## Antimycin A treatment

Samples were prepared and placed on the microscope stage. Immediately after imaging for the first time (time point 0 h), 100 μM antimycin A (Sigma-Aldrich, #A8674) was added.

## Immunofluorescence

Wing discs from up-crawling third-instar larvae were stained as described earlier (Greco *et al*, 2001). Briefly, wing discs were dissected in PBS, fixed in 4% paraformaldehyde for 20 min and rinsed five times in PBS. Wing discs were then permeabilized with 0.05% Triton X-100 in PBS (PBX) twice for 10 min, blocked three times for 15 min in PBX + 1 mg/ml BSA + 250 mM NaCl and incubated overnight with the primary antibody in PBX + 1 mg/ml BSA (BBX) at 4°C. After washing twice for 20 min in BBX, wing discs were blocked twice for 20 min in the blocking solution BBX + 4% normal goat serum and incubated for at least 3 h with the secondary antibody in the blocking solution. After washing three times for 15 min each in PBX and twice in PBS, wing discs were finally mounted in VectaShield® (Vector Labs, #H-1000). To stain ring glands, ring glands from up-crawling third-instar larvae were dissected and fixed as wing discs but were then permeabilized with 0.1% Triton X-100 in PBS three times for 10 min, blocked for 30 min – 1 h in 0.1% Triton X-100 in PBS + 10% normal goat serum and incubated overnight with the primary antibody in the blocking solution at 4°C. After washing three times for 10 min in blocking solution, ring glands were incubated for 3 h with the secondary antibody in the blocking solution. After washing twice for 10 min with 0.1% Triton X-100 in PBS and once with PBS, ring glands were finally mounted in ProLong Gold (Thermo Fisher Scientific, #P10144). The following primary antibodies were used: rabbit anti-

cleaved Caspase-3 (Asp175) (1:500) (Cell Signaling #9661), mouse anti-Dlg (1:200) (DSHB, #4F3, concentrate), chicken anti-ß Galactosidase (1:2,000) (Abcam, #ab134435), rat anti-Ci$_{155}$ (1:30) (DSHB, #2A1, concentrate), rat anti-Crb2.8 (1:1,000) (Richard *et al*, 2006) (a gift from E. Knust, MPI-CBG, Dresden, Germany), chicken anti-GFP (1:1,000) (Abcam, #ab13970), rabbit anti-GFP (1:1,500) (Thermo Fisher Scientific, #A-11122), rabbit anti-Hh (1:500) (Richard *et al*, 2006) and mouse anti-Smo (1:50) (DSHB, #20C6, concentrate). Secondary antibodies conjugated with Alexa Fluor® 488, 555 and 647 were diluted 1:1,000 or, in the latter case, 1:500 (Thermo Fisher Scientific).

NIH3T3/Smo-mEos2 cells treated with SAG, 2-DG or 3-BP were rinsed with PBS a few times before fixing with 4% paraformaldehyde for 10 min at room temperature. After washing twice with PBS, cells were permeabilized with 0.15% Triton X-100, 5% normal goat serum and 0.1% BSA in PBS for 20 min at room temperature. Then, cells were incubated overnight at 4°C with anti-acetylated tubulin (1:1,000) (Sigma-Aldrich, #T7451) as primary antibody to stain for primary cilia. After washing several times with PBS, cells were incubated with goat anti-mouse Alexa Fluor® 647 (1:1,000) (Thermo Fisher Scientific, #A-21236) as secondary antibody for 1 h at room temperature. Cells were washed again before staining with DAPI (1:10,000) (Roche, #10236276001) for 10 min at room temperature and then mounted onto glass slides in VectaShield®.

## smFISH

Custom Stellaris® FISH probes were designed against *GFP* or *smo* mRNA by utilizing the Stellaris® FISH probe designer Biosearch Technologies, Inc. (Petaluma, CA, USA). Wing discs were hybridized with *GFP* and *smo* Stellaris® FISH probe sets labelled with Quasar 670 and Quasar 570 (Biosearch Technologies, Inc.), respectively, following the manufacture's protocol using 250 nM per probe set. Briefly, wing discs from up-crawling third-instar larvae were dissected in PBS, fixed with 4% paraformaldehyde for 40 min on ice and then washed twice in PBS for 5 min. For permeabilization, wing discs were incubated overnight in 70% ethanol at 4°C. After washing for 10 min with wash buffer A (Biosearch Technologies, #SMF-WA1-60), wing discs were incubated overnight with Stellaris® FISH probes in hybridization buffer (Biosearch Technologies, #SMF-HB1-10) at 37°C. Subsequently, wing discs were washed in wash buffer A for 30 min at 37°C and in wash buffer B (Biosearch Technologies, #SMF-WA1-20) for 30 min at room temperature. Wing discs were finally mounted in VectaShield®.

## RNA extraction and cDNA synthesis

For each genotype, 50 wing discs were dissected in ice-cold PBS from up-crawling third-instar larvae and collected in an iced 1.5-ml microcentrifuge tube. PBS was removed as much as possible before proceeding immediately with total RNA extraction using the RNeasy Mini Kit (Qiagen, #74104) according to the manufacturer's protocol (including on-column DNase digestion). Then, 500 ng of total RNA was used to synthesize cDNA from polyadenylated RNA using SuperScript® III Reverse Transcriptase Kit (Thermo Fisher Scientific, #18080093) according to the manufacturer's protocol.

## Semi-quantitative PCR

PCR using Phusion High-Fidelity DNA Polymerase (NEB, #M0530L) was performed according to standard protocols (27 cycles) with 2 µl of cDNA, using water as a negative control and genomic DNA from wild-type flies (50–75 ng/µl) as a positive control. The intensities of the PCR bands were quantified using the Gels plugin in Fiji (Schindelin *et al*, 2012). The following primer pairs were used (see Appendix Table S3 for sequence information): eIF4A-F and eIF4A-R, eIF4A_intron_f and eIF4A_intron_r, Gapdh2-F and Gapdh2-R, Gapdh2-intron-F and Gapdh2-intron-R or Gapdh2-intron-R2, Pdk-F and Pdk-R, Pdk-intron-F and Pdk-intron-R, PyK-F and PyK-R, PyK-intron-F and PyK-R, smo-F and smo-R, and smo-intron-F and smo-intron-R.

## Protein extracts from wing imaginal discs

For each genotype, 25 wing discs were dissected in ice-cold PBS from up-crawling third-instar larvae and collected in an iced 1.5-ml microcentrifuge tube. PBS was removed as much as possible before adding 25 µl RIPA buffer (50 mM Tris–HCl pH 7.5, 150 mM NaCl, 0.1% SDS, 1% sodium deoxycholate, 1% Triton X-100, 1% NP-40). Samples were homogenized on ice using a BioVortexer. SDS loading buffer was added to the samples before analysed by Western blotting.

## Western blotting

Western blotting was essentially performed according to standard protocols. The primary antibodies used were as follows: goat anti-E-Cad dP-20 (1:100) (Santa Cruz Biotechnology, #sc-15751) and guinea pig anti-Ecd C-term (1:750) (this study). The secondary antibodies used were as follows: anti-goat HRP (1:5,000) (Millipore, #AP180P) and anti-guinea pig HRP (1:2,500) (Dianova, #106-035-003).

## Membrane potential assay

To investigate the plasma membrane potential, we used the membrane potential-sensitive dye DiBAC$_4$(3) (Thermo Fisher Scientific, #B438). Wing discs from up-crawling third-instar larvae were dissected in Grace's medium supplemented with 5% FBS and 20 nM 20-hydroxyecdysone (full medium) (Dye *et al*, 2017), and incubated with 500 nM DiBAC$_4$(3) in full medium for 15 min at 29°C. After washing twice with full medium, wing discs were mounted and imaged immediately. A higher fluorescence intensity of DiBAC$_4$(3) indicates depolarization of the plasma membrane.

## PAC-NAE synthesis

PAC-NAE was synthesized as follows. A solution of PAC-FA (100 mg, 378 µmol) (Haberkant *et al*, 2013), HBTU (100 mg, 267 µmol), HOBT (10 mg, 74 µmol) and DIEA (100 µl, 575 µmol) in NMP (3 ml) was treated with ethanolamine (300 µl, 4.97 mmol) and stirred for 1 h. The reaction mixture was transferred onto a mixture of ethyl acetate and H$_2$O (1:1, 200 ml). The layers were separated and the organic layer washed with H$_2$O (3 × 100 ml) and saturated NaCl solution (100 ml) and dried over MgSO$_4$. The solvent was removed under reduced pressure and the residue purified by

flash chromatography on silica gel using the eluent system dichloromethane/MeOH 95:5. The product was obtained as a colourless solid.

Chemical Formula: $C_{17}H_{29}N_3O_2$
Exact Mass: 307.23
Molecular Weight: 307.44

[1]H-NMR (400 MHz, CDCl3) δ = δ 6.03 (s, 1H), 3.76–3.68 (t, J = 5.3 Hz, 2H), 3.41 (m, 2H), 2.25–2.12 (m, 4H), 1.95 (t, J = 2.7 Hz, 1H), 1.62 (m, 2H), 1.48 (m, 2H), 1.40–1.17 (m, 12H), 1.06 (m, 2H) ppm.

[13]C-NMR (100 MHz, CDCl3) δ = 174.58, 83.46, 68.88, 62.62, 42.55, 36.56, 32.82, 31.84, 29.17, 29.15, 29.12, 29.09, 28.44, 25.65, 23.77, 22.75, 17.97 ppm.

MS (ESI) calculated for $[M+H^+]$: 308.23, found: 308.23

Yield: 52 mg (169 μmol, 45%)

## Lipophorin isolation and labelling with PAC-NAE

The isolation procedure of Lipophorin (Lpp) particles was adapted from Panakova *et al* (2005). Briefly, 10 ml feeding third-instar larvae were first washed with water and then with TNE buffer (100 mM Tris–HCl pH 7.5, 150 mM NaCl, 0.2 mM EGTA, cOmplete™ protease inhibitor cocktail (Roche, #0000000011697498001)). All following steps were conducted at 4 °C (unless stated otherwise). Larvae were homogenized in TNE buffer in a tissue grinder with a loose pestle, and the homogenate was centrifuged for 10 min at 1,000 × *g* and afterwards for 3 h at 142,000 × *g*. The resulting supernatant (10 ml) was mixed with 40 μl 10 mM PAC-NAE (EtOH solution) and sonicated for 5 min. Subsequently, KBr was added to a final concentration of 0.33 mg/ml, and the samples were centrifuged at 192,000 × *g* for 64 h at 8°C. The top fraction (containing labelled Lpp particles) was desalted on NAP-5 columns (Sigma-Aldrich, #GE17-0853-01). Lpp particles were eluted with Grace's medium, aliquoted, snap-frozen and stored at −80°C.

The concentration of PAC-NAE on Lpp particles was determined by a combination of two-step lipid extraction, click chemistry and TLC. Briefly, lipids were extracted from Lpp by a two-step procedure. The volume of Lpp particles was brought to 100 μl with PBS and mixed with 100 μl chloroform:methanol 10:1 (v/v) for 5 min in an Eppendorf ThermoMixer (1200 rpm). Afterwards, samples were centrifuged at 14,000 × *g* for 5 min and the lower (organic) phase was collected to a new tube. Next, the upper (hydrophilic) phase was re-extracted with 100 μl chloroform:methanol 2:1 (v/v). The pooled lipid fractions were dried in a SpeedVac, mixed with 30 μl click reaction mixture (0.27 μl 10 mM 3-azido-7-hydroxycoumarin, 6.75 μl 10 mM $[acetonitrile]_4CuBF_4$, 22.98 μl EtOH), shortly vortexed and incubated for 30 min in a SpeedVac at 35-40 °C (until the complete evaporation of the solvent). Afterwards, lipid extracts were dissolved in chloroform:methanol 2:1 (v/v) and 10 μl was applied onto a 10 × 10 cm standard silica TLC plate (Merck, #105633). The plate was developed in $CHCl_3$:MeOH:$H_2O$:AcOH 65:25:4:1 (v/v/v/v) for 5 cm, dried in a stream of warm air and further developed in hexane:ethyl acetate 1:1 (v/v) (9 cm) (adapted from Gaebler *et al* (2013)). Imaging was performed with a GelDoc

system (365 nm). In parallel, the standard curve of PAC-NAE was prepared and used for calculation of the concentration. The final concentration of PAC-NAE was within 1–2 μM range.

## Lipophorin-mediated PAC-NAE uptake

Wing discs from up-crawling third-instar larvae were dissected in Grace's medium, washed twice and incubated for 1 min with 1.5 μM PAC-NAE labelled Lpp particles. Afterwards, wing discs were washed with PBS and fixed with 4% paraformaldehyde for 20 min at room temperature. After washing twice with 0.05% Triton X in PBS for 10 min, the conjugation between the alkyne (PAC-NAE) and azide groups (Alexa Fluor 555 azide, Thermo Fisher Scientific, #A20012) was performed via click chemistry reaction using Click-iT Cell Reaction Buffer Kit (Thermo Fisher Scientific, #C10269). Subsequently, wing discs were mounted in VectaShield® and imaged.

PAC-NAE uptake after gramicidin A treatment was performed as described above, but wing discs were chased with Grace's medium supplemented with 5% FBS and 20 nM 20-hydroxyecdysone (Dye *et al*, 2017) for 20 min before fixation and click chemistry reaction.

## Gramicidin A treatment

Wild-type wing discs from up-crawling third-instar larvae were dissected in Grace's medium supplemented with 5% FBS and 20 nM 20-hydroxyecdysone (full medium) (Dye *et al*, 2017) and incubated with 1 μM gramicidin A (Sigma-Aldrich, #50845) in full medium for 30 min at 29°C. Before proceeding with either the membrane potential assay or PAC-NAE uptake, wing discs were washed twice with full medium.

## Mounting of adult wings

Male flies were collected in isopropanol 2 days after eclosion. After at least 2 days in isopropanol, wings were dissected in isopropanol and mounted on glass slides using the resin Euparal (Carl-Roth, #7356.2) as mounting medium.

## Microscopy and image analysis

Immunohistochemical images were acquired using a Zeiss LSM 510 confocal microscope equipped with a Plan-Neofluar 40×/1.3 oil objective, an Olympus FluoViewTM FV1000 confocal microscope equipped with an UPlanSApochromat 60×/1.35 oil objective, or using a Zeiss LSM 880 confocal microscope equipped with a Zeiss C-Apochromat 40×/1.2 W or Zeiss Plan-Apochromat 63x/1.4 Oil DIC objective. Images of adult wings were acquired using a Zeiss Axioscan.Z1 widefield slide scanner equipped with a Plan-Apochromat 10×/0.45 air objective as a series of tiles with 10% overlap that were stitched together by the microscope software (convex hull tile module, online stitching method, Zeiss ZEN). Fiji (Schindelin *et al*, 2012) was used for image processing and analysis. All images of wing discs were orientated so that the A/P boundary is vertical and the D/V boundary is horizontal, with the anterior compartment to the left and the dorsal compartment to the bottom. Immunohistochemical images show maximal projections that were generated using the maximal intensity function in Fiji.

Only ring glands and wing discs that were processed in parallel, imaged using the same microscope settings, and properly mounted were used for quantifications. For quantification of immunohistochemical staining of CD8::GFP in ring glands, the mean pixel intensity of two regions of interest of $25 \times 25$ μm per ring gland was determined using ROI manager, and the average of these was used for the statistical analysis. For quantification of compartment size, wing disc folds served as morphological landmarks to outline the region of the wing pouch and the D/V boundary. The area of dorsal and ventral compartment was then determined using ROI manager. For quantification of immunohistochemical staining for Cas3*, regions of interest ($87.5 \times 22.5$ μm each) were selected in the dorsal and ventral compartments and the mean pixel intensity in these regions was determined using ROI manager. For quantification of immunohistochemical stainings for $Ci_{155}$, *dppZ*, Hh and Smo, a rectangle of $100 \times 20$ μm was positioned in a way that 80 μm of its width covered the anterior compartment, while the remaining 20 μm of its width covered the posterior compartment. The A/P boundary was determined according to anti-$Ci_{155}$ or anti-Ptc co-immunostaining. The pixel intensity was determined as a function of distance from the A/P boundary using the Plot Profile function in FIJI. For quantification of PAC-NAE pixel intensity upon gramicidin A treatment, a region of interest of $103.5 \times 74.5$ μm was selected. The mean grey value was measured using ROI manager. PAC-NAE pixel intensity upon genetic perturbations was quantified the same way, but regions of interest of $80 \times 22$ μm and $52 \times 22$ μm were selected anteriorly and posteriorly, respectively, in the dorsal and ventral compartments. Pixel intensity of each compartment is plotted as the average of the anterior and posterior mean grey value. For quantification of $DiBAC_4(3)$ pixel intensity, *z*-stacks of wing discs were resliced, avoiding interpolation from anterior to posterior. The sum of slices corresponding to 104 μm total thickness was generated. For the analysis of the effect of genetic perturbations, a region of interest of 41.5 μm x the size of the *z*-stack was selected in the dorsal and ventral compartments, respectively. For the analysis of the effect of gramicidin A, a region of interest of 83 μm x the size of the z-stack was selected. The integrated density was measured using ROI manager. For quantification of smFISH spots, regions of interest ($75 \times 26$ μm each) were selected in the dorsal and ventral compartments. After setting the threshold manually, smFISH spots were counted using the 3D Object Counter function. To determine the number of cilia and calculate the percentage of Smo localization in NIH3T3/Smo-mEos2 cells, images from three technical replicates were analysed using Fiji. The size and shape of adult wings were measured using a custom-made Fiji macro that calculates the major and minor axes, as well as the total area of the wing blade based on manual delimitation of the perimeter of the wing blade. Pixel calculations were converted into millimetres.

For all quantifications, GraphPad Prism 5 was used to plot the results and to perform the statistical analyses. Statistical details of experiments can be found in the figures and figure legends.

### Statistical analysis of $Ci_{155}$, *dppZ*, Hh and Smo expression

To determine whether the expression of $Ci_{155}$, *dppZ*, Hh and Smo is significantly affected after the different genetic perturbations used in this study, the area under the curve of the representative staining was determined for each wing disc. To correct for differences in staining intensity between wing discs, the ratio of the area under the curve of the representative staining in the dorsal and ventral compartments was calculated for each perturbation and compared to the ratio from control wing discs using Student's *t*-test. For $enGal^{ts} > ecd^{RNAi}$ and $dppGal^{ts} > ecd^{RNAi}$ experiments, differences in staining intensity between wing discs were corrected by adjusting the pixel intensity of control wing discs to match pixel intensity of $enGal^{ts} > ecd^{RNAi}$ or $dppGal^{ts} > ecd^{RNAi}$ wing discs in the posterior compartment. Hence, the areas under the curve of the representative staining in control and $enGal^{ts} > ecd^{RNAi}$ or $dppGal^{ts} > ecd^{RNAi}$ wing discs could be compared using Student's *t*-test.

**Expanded View** for this article is available online.

## Acknowledgements
This paper is dedicated to the memory of our wonderful colleague, Prof. Dr. Suzanne Eaton, who sadly passed away on 2 July 2019. We thank BDSC and VDRC for fly stocks and DSHB for antibodies; Elisabeth Knust and Tadashi Uemura for sharing reagents; Philip Beachy for providing NIH3T3/Smo-mEos2 cells; Thomas Hurd for hosting S.S. and providing resources during the revision process; the Light Microscopy Facility, scientific computing facility and fly resource team at MPI-CBG for excellent support; the Light Microscopy Facility of the CMCB Technology Platform at TU Dresden; Ali Mahmoud for his great laboratory management and technical assistance; Petra Born for her assistance in generating preliminary data in NIH3T3/Smo-mEos2 cells; Andreas Sagner, Tom Kazimiers and Franz Gruber for writing a FIJI macro to analyse wing size and shape; Sebastian Dunst for critical discussions; and Mark Leaver, Kate Lee and Avinash Patel for reading and commenting on the manuscript. This work was supported by the DFG (SFB/TRR 83) and the Max Planck Society. Open access funding enabled and organized by Projekt DEAL.

## Conflict of interest
The authors declare that they have no conflict of interest.

## Author contributions
SS performed the experiments in Figs 2, 3 and 5, and EV1–EV5, and Appendix Figs S1 and S3–S6. TB generated the *Gapdh1::GFP* and *GFP::Gapdh2* CRISPR/Cas9 lines and performed the experiments in Fig 6. IN performed the experiments in Fig 1, with methods established by KVI, and Appendix Fig S2, with contribution by NAD. SAZ performed the experiments in Fig 4. HK performed key preliminary experiments. AN synthesized PAC-NAE in consultation with CS. SS and SE designed the study. SS, NAD and SE wrote the manuscript with discussions and feedback from all authors.

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
