## [Review Process File · The EMBO Journal]

Glycolysis regulates Hedgehog signaling via the plasma membrane potential

Stephanie Spannll, Tomasz Buhl, Ioannis Nellas, Salma Zeidan, K. Iyer, Helena Khaliullina, Carsten Schultz, André Nadler, Natalie Dye, and Suzanne Eaton*

* Deceased July 2, 2019

DOI: [10.15252/embj.2019101767](https://doi.org/10.15252/embj.2019101767)

Corresponding authors: Natalie Dye (dye@mpi-cbg.de) , Stephanie Spannll (steffi.spannll@utoronto.ca)

Review Timeline:	Submission Date:	11th Feb 19
	Editorial Decision:	8th Apr 19
	Revision Received:	10th Jun 20
	Editorial Decision:	14th Jul 20
	Revision Received:	19th Aug 20
	Accepted:	25th Aug 20

Editor: Daniel Klimmeck

Transaction Report:

Dear Suzanne,

Thank you again for the submission of your manuscript (EMBOJ-2019-101767) to The EMBO Journal and in addition providing us with a preliminary revision plan, and also for your patience with my response, which got delayed due to detailed discussions in the team regarding your point-by-point response. As mentioned earlier, your study has been sent to three referees, and we have received reports from all of them, which I enclose below.

The referees acknowledge the potential interest and novelty of your work, although they also express major concerns. In particular, referee #2 raises issues regarding Ecdysoneless as a model to study glycolytic dependences of wing disc signaling and states that potential indirect confounding effects significantly weaken the impact of the current results in his-her view. Referee #3 agrees in that the physiological relevance of the data is not sufficiently worked out at this stage and additional experimentation would be needed to corroborate a function for Ecd-glycolysis in normal hh signaling. Further, referee #2 finds cross-dependencies on cell proliferation control and the mechanistic details linking ecd to membrane potential are not sufficiently investigated. In addition, the referees point to issues related to missing controls and statistics that are not at the level of completeness required for publication at The EMBO Journal.

I judge the comments of the referees to be generally reasonable and given their overall interest, we are in principle happy to invite you to revise your manuscript experimentally to address the referees' comments, along the lines sketched in your outline. We concur in particular with the referees that the proof of relevance for an explicit glycolytic program should be a key consideration for this study. I need to stress though that we do need strong support from the referees on a revised version of the study in order to move on to publication of the work.

Please feel free to contact me if you have any questions or need further input on the referee comments.

Thank you for the opportunity to consider your work for publication. I look forward to your revision.

Kind regards,

Daniel

Daniel Klimmeck, PhD
Editor
The EMBO Journal

http://embopress.org/sites/default/files/EMBOPress_Figure_Guidelines_061115.pdf

- a point-by-point response to the referees' comments, with a detailed description of the changes made (as a word file).
- a word file of the manuscript text.
- individual production quality figure files (one file per figure)
- a complete author checklist, which you can download from our author guidelines (<http://emboj.embopress.org/authorguide>).
- Expanded View files (replacing Supplementary Information)

<http://emboj.embopress.org/authorguide#expandedview>

Further information is available in our Guide For Authors:

http://emboj.msubmit.net/html/emboj_author_instructions.html

The revision must be submitted online within 90 days; please click on the link below to submit the revision online before 7th Jul 2019.

Link Not Available

Referee #1:

Spannl and colleagues nicely demonstrated in this paper that glycolysis is involved in regulating hedgehog signalling in wing imaginal discs. Reduction of glycolysis stabilizes smoothed and the full-length Cubitus interruptus, a transcription activator of hedgehog signalling pathway.

Mechanistically, this is achieved by lowering the plasma membrane potential, which is required for the uptake of N- acylethanolamides, a hedgehog pathway inhibitor. The data in this paper are of high quality, and the claims are well substantiated, furthermore, the ms is clearly written. This major claims in the paper are highly novel, since it reveals interplay between cell metabolism and signalling pathway important for proliferation and differentiation. However, I am not completely convinced by the claim that decreasing Ecdysoneless promotes hedgehog signalling pathway via reducing glycolysis. Clarification of the following points would be most helpful.

Major issues:

1. Line 106 -126, the authors claim that Ecd knockdown reduces the ATP levels via lowering glycolysis, as Ecd knockdown leads to reduction of glycolysis- promoting enzymes. As it was shown previously, that Ecd mutation causes disruption of the production of ecdysone, which is involved in growth regulation, and the authors showed themselves, the Ecd RNAi caused reduction in growth Figure 1E; how do the authors know that reduction of ATP is not a result of decreased cell proliferation? Are there mutants where growth is affected, but ATP levels are not?

2. From Figure 2, and line 128-151, apGalts > ecdRNAi and apGalts > GapdhRNAi have similar strength in stabilizing Smo. However, in Figure 4, apGalts > GapdhRNAi seems to have weaker effect on PAC-NAE absorbance as well as plasma membrane potential than apGalts > ecdRNAi. Could there be a separate mechanism how ecd regulate plasma membrane potential that is independent of Gapdh?

3. In Figure 2 B', C' and E, from the images provided, it is apparent that there is an upregulation of smo in the dorsal compartment, and this is supported by the quantifications. However, in B', C' and E', though the quantifications demonstrate an increase of Ci in the dorsal vs ventral compartment (comparable to that of smo), the figures B' C' and E' really doesn't look very obviously different between the dorsal vs ventral compartments. Perhaps some more convincing images are required.

Minor issues:

In some of the EV figures, e.g. EV3, the wing disc images were accompanied with a schematic of a wing disc, with a/p and v/d domains depicted, and clear depiction of where the drivers are expressed. I have found these very useful, and it would aid the readers greatly, if these schematics can be made for all the wing disc figures.

Referee #2:

Cell metabolism influences Hedgehog signaling in the Drosophila wing disc

In this study the authors asked how cell metabolism and plasma membrane potential are linked to Hedgehog (Hh) signaling pathway in wing disc development. Hh itself and some of the members of the Hh signaling cascade were previously shown to be regulated by addition of metabolites, such as lipids, which makes this signaling pathway a great candidate to be the mediator of the interplay between metabolism and developmental patterning.

The authors perform this study in the wing disc and use apGalts which allows them to nicely interfere with candidate genes and simultaneously have an internal control for the study of gene expression, tissue growth and patterning.

The authors have used ecdysoneless (Ecd), a protein with unclear function, as a tool to affect glycolysis and found that somehow Ecd affects ATP levels in cells of the imaginal wing disc, and

that it regulates splicing of some glycolytic enzymes. Ecd knock down was also shown to lead to increased levels of Smo and Ci155 independent of Hh release. To test if this increase in Smo and Ci was a consequence of Ecd role in glycolysis regulation, the authors have directly knocked down Gapdh2, a glycolytic enzyme. Gapdh2 RNAi causes a similar accumulation of Smo and Ci155. The effect of glycolysis levels on Smo localization was nicely shown to also occur in mammalian cells. The authors have then explored how glycolysis could be affecting Smo and Ci155 and found that RNAi of Ecd or Gapdh lowers plasma membrane potential and also leads to a reduction in the accumulation of an analogue of N-acylethanolamide (PAC-NAE) and that these two aspects are inter-related.

Major comments:

The authors explored a very interesting question of how cellular metabolism and signaling pathways involved in patterning are inter-regulated.

To answer this question the authors choose to analyse Ecdysoneless, a protein with unknown function. Ecdysoneless was identified and named after its phenotype which is very similar to what is caused by deficient Ecdysone signaling, however its exact function remains unclear. To use a protein like Ecd to alter glycolysis is in my opinion not ideal, because it is not known what EcdIR is causing at a mechanistic level and since glycolysis is very well characterized the authors could have chosen many other ways to affect it in a cleaner way.

As previously reported for Ecd in the PG, the authors now find that Ecd in the wing disc somehow regulates splicing and that it specifically affects splicing of Gapdh, Pdk and PyK. This defect could serve as an explanation for how Ecd affects glycolysis levels. It remains unanswered if these splicing defects indeed lead to a decrease in glycolysis (for instance with measurement of lactate), or if Ecd also affects splicing of other metabolic enzymes (for instance in Krebs cycle).

The authors also directly try to affect glycolysis by knocking down Gapdh2 (again no lactate direct measurement) and find that both Smo and Ci155 are affected in a similar manner as in EcdIR. The authors analysis of the role of Hh release in the accumulation of Smo and Ci155 in the background of EcdIR or GapdhIR (in Dispatched mutant background) is not very convincing since the increase in both Smo and Ci155 in these backgrounds is very small and may not be significant. The decrease in plasma membrane potential and decreased accumulation of PAC-NAE in both EcdIR or GapdhIR are very interesting observations. It would be important to confirm if this is specific for Gapdh or if affecting glycolysis in other ways (or other metabolic pathways) also causes this same phenotype. It would also be important to test if disrupting membrane potential also leads to an accumulation of Smo and Ci155 and therefore this is the mechanism by which reduction in Gapdh (and potentially glycolysis levels) is interfering with the distribution/levels of these proteins. Overall this manuscript addresses a very interesting question, it has a series of very interesting experiments/data but in the current format it is hard to conceptually integrate them all.

Specific comments:

- 1- 92 - even if Ecd has been shown to not be involved in steroid hormone production it could still be involved in ecdysone signaling.
- 2- Fig. 1B' and C' - Caspase 3 quantification missing.
- 3- Fig. EV1 - Is Ecd a regulator of splicing? Would you see the same phenotype if you would knocked down a bone-fide splicing regulator (e.g Nelf-a)?
- 4- Fig. EV1 - Is this role of Ecd in splicing specific for the enzymes tested? Did you look at other glycolysis enzymes or to Krebs cycle enzymes? If Ecd affects splicing of multiple enzymes in multiple pathways the overall effect in metabolism can be unpredictable.

5- Lower ATP levels do not necessarily mean lower glycolysis; lower levels of spliced Gapdh, Pyk, etc do not mean lower glycolysis - Quantify glycolysis (e.g. lactate chemical measurement).

6- The authors mention that Ecd mutants can be rescued when UAS-Ecd x Ptc-Gal4 - Could Ecd be also required for the splicing of Dpp?

The experiment with dpp-LacZ does not answer this question.

Note: In ecd2 or if Dpp is not correctly made are discs smaller? Smaller discs are known to prevent timely molting.

7- 154 - "glycolysis on Smo trafficking" the effect on trafficking was not really shown in Drosophila. The accumulation of Smo could have many causes. Are total levels of Smo in Drosophila EcdIR/Gapdh and in mammalian cells treated with 2-DG the same?

8- 165 - "trafficking" - not shown.

9- Fig 3C-F - in the absence of Disp, loss of Ecd does not cause the same accumulation of Smo or Ci as in the presence of Disp. This difference could indeed mean that Hh release is required. How can this decrease be otherwise explained?

10- 183 - Lpp = meaning? HhN=meaning?

11- Fig. EV1 H' - Show area of gel which would show the unspliced form of eIF-4a and rp49 (as in G' or F').

12- Fig. EV1 H' - Why is there no band in gel in DNA lane for eIF4a? Why is the band of rp49 in DNA lane heavier?

Minor comments:

Title: It is too general. A title more directed to the results of the manuscript would be better.

Abstract: Line 25 - "We show that reducing glycolysis" - the authors have never shown this.

Reducing the levels of ATP can have many causes. It is also not clear if the tools the authors use indeed lead to a reduction in glycolysis.

Introduction: Line 42 - remove "respectively". A-KG could both affect methylation and acetylation.

Line 53 - "Smo can be activated by cholesterol..." in vertebrates.

Referee #3:

In this manuscript, the authors investigate the role of metabolic reprogramming in the regulation of Hedgehog signaling within the Drosophila wing disc. First, the authors present evidence that inhibition of glycolysis stabilizes smoothened and ci in the dorsal compartment. The effects are quite subtle, but they observe consistent results when glycolysis is inhibited by multiple means (RNAi knockdown of ecdysoneless, which is a splicing factor with conserved roles in regulating metabolism, or RNAi against the metabolic enzymes Gapdh2 or Glo1). Next, they provide evidence that RNAi knockdown of Ecd or Gapdh2 is associated with lower membrane potential and reduced uptake of N-acylethanolamides, though again the effects are small. Overall, their study points to a model in which reduced glycolysis inhibits membrane production, which limits uptake of N-acylethanolamides. This, in turn, stabilizes smoothened protein, thus mimicking the effect of patched inhibition in a ligand-independent manner. The connection between metabolism and cell signaling has been understudied, so investigations of this type have the potential to break new ground. The strengths of this study include the use of multiple independent methods to test for the effects of decreased glycolysis, clear images and careful quantification of the observations, and testing for effects in both Drosophila tissues and mammalian cells. With these strengths, I think the study has the potential to make a significant contribution to the field. However, several comments should be addressed before publication.

1. First, my enthusiasm for this study was diminished by the very small size of the effects in nearly

every case (Fig. 1E & G, 2B-E, 3D & F, and 4A). Although these differences are shown to be statistically significant (in the case of Fig. 1 and 4), it is not clear that these differences are biologically meaningful. The authors do show that one biologically relevant output, dpp expression, is affected (Fig. 3H & J), but additional evidence along these lines would be helpful. Alternatively (or in addition), it would be interesting to determine whether differences in Ecd expression and/or the rate of glycolysis in contributes to the normal patterning of Hh signaling in wildtype tissue in a meaningful way. A third possibility would be to assay for more dramatic effects on Hh signaling or biological outcome in homozygous mutant clones. Although it may not be necessary to pursue all three of these lines of inquiry, I think that some additional data about either the biological impact of the experimental perturbations that have already been used in this study or the ways in which changes in metabolism within wildtype tissue contributes to the patterning of Hh signaling is important and would significantly enhance the impact of the study.

2. In Figures 2 and 3, the standard deviations of the red and black lines overlap in most cases, raising the possibility that the differences between the means are not statistically significant. A statistical procedure to should be used to test whether these data sets are significantly different from each other.

3. In Fig. 3I, the relevant parts of the image (the primary cilia) are located around the periphery of the image and are almost obscured by the image label text. An alternate image or layout should be used so that it is easier to see the effect.

RESPONSE TO REVIEWER COMMENTS:

Spannl et al EMBOJ-2019-101767

Referee #1:**Major issues:**

1. Line 106 -126, the authors claim that Ecd knockdown reduces the ATP levels via lowering glycolysis, as Ecd knockdown leads to reduction of glycolysis- promoting enzymes. As it was shown previously, that Ecd mutation causes disruption of the production of ecdysone, which is involved in growth regulation, and the authors showed themselves, the Ecd RNAi caused reduction in growth Figure 1E; how do the authors know that reduction of ATP is not a result of decreased cell proliferation? Are there mutants where growth is affected, but ATP levels are not?

While it is true that whole animals mutant for Ecd produce less ecdysone, this fact is not relevant for our experiments because we specifically knocked down *ecd* in the dorsal compartment of the wing disc using *ap-Gal4*. We added data (Fig EV1A-C) confirming that *ap-Gal4* does not drive expression in the ecdysone-producing ring gland. Thus, circulating ecdysone levels should be normal, and the growth defect we see in the wing is autonomously caused by loss of Ecd.

With respect to ATP levels and growth rate, we are not aware of any evidence that slowing tissue growth lowers steady state ATP levels in any system – in bacteria, there is no change in steady state ATP concentration between exponential and stationary phase (Schneider and Gourse, JBC, 2004). In fact, unpublished work from our lab indicates that overgrowth caused by activation of the insulin receptor is actually associated with lowered steady state ATP levels.

2. From Figure 2, and line 128-151, *apGalts > ecdRNAi* and *apGalts > GapdhRNAi* have similar strength in stabilizing Smo. However, in Figure 4, *apGalts > GapdhRNAi* seems to have weaker effect on PAC-NAE absorbance as well as plasma membrane potential than *apGalts > ecdRNAi*. Could there be a separate mechanism how *ecd* regulate plasma membrane potential that is independent of Gapdh?

The effect of *ecd^{RNAi}* on Smo accumulation does appear to depend on its role as a splicing regulator. We have found that knock-down of *brr2* (the splicing factor with which Ecd interacts) causes similar changes in Smo (new data included in Fig EV4C-C''). Since Ecd seems to regulate splicing of other glycolytic enzymes in addition to Gapdh, it might be expected to cause a stronger effect on membrane potential and NAE uptake than perturbing Gapdh alone. We have already ruled out a simple effect of Ecd on *smo* splicing, although it is also possible that Ecd exerts its effects on Smo through splicing of non-metabolic RNAs. Nonetheless, the fact that loss of Gapdh produces a similar (though milder) phenotype suggests that changes in cell metabolism would be sufficient to explain the effect of *ecd^{RNAi}* on Smo.

3. In Figure 2 B', C' and E, from the images provided, it is apparent that there is an upregulation of *smo* in the dorsal compartment, and this is supported by the quantifications. However, in B', C' and E', though the quantifications demonstrate an increase of Ci in the dorsal vs ventral compartment (comparable to that of *smo*), the figures B' C' and E' really doesn't look very obviously different between the dorsal vs ventral compartments. Perhaps some more convincing images are required.

These data are now in Fig 3. We have adjusted the brightness/contrast to improve visualization of the phenotype. Additionally, our quantification of multiple discs shows the effect on Ci_{155} is significant. Perhaps what this reviewer is noticing is that Ci_{155} does not accumulate over the same range as Smo. While Smo levels increase uniformly throughout the anterior compartment, the effect on Ci_{155} is not strong in far anterior regions of the disc. We have changed the text to clarify this point (pg 10, lines 214).

This difference in Ci_{155} might reflect the fact that processing of Ci_{155} is regulated by multiple mechanisms. Smo membrane accumulation upon loss of Ecd might only affect some but not all of these mechanisms. Examples of such mechanisms are: the Ci-binding protein, Debra, which in the anterior compartment is highly expressed in a band of 6-8 cells adjacent to the Hh-responsive cells that express Dpp, mediates polyubiquitination and lysosomal degradation of Ci_{155} (Dai et al., 2003). Further, Ci_{155} stability in more anterior cells is controlled by the Slimb-Cul1 E3 ligase that mediates

ubiquitination and partial degradation of Ci₁₅₅ to its repressor form Ci₇₅ (Aza-Blanc et al., 1997; Jiang, 2006; Zhang et al., 2013). Further investigation into how loss of Ecd leads to Ci₁₅₅ stabilization is beyond the scope of the current manuscript.

Minor issues:

In some of the EV figures, e.g. EV3, the wing disc images were accompanied with a schematic of a wing disc, with a/p and v/d domains depicted, and clear depiction of where the drivers are expressed. I have found these very useful, and it would aid the readers greatly, if these schematics can be made for all the wing disc figures.

Due to space limitations, we would prefer not to add this schematic to all relevant figures. However, we have instead added a reference to Fig 1C in the figure legends for all other figures.

Referee #2:

Major comments:

The authors explored a very interesting question of how cellular metabolism and signaling pathways involved in patterning are inter-regulated. To answer this question the authors choose to analyse Ecdysoneless, a protein with unknown function. Ecdysoneless was identified and named after its phenotype which is very similar to what is caused by deficient Ecdysone signaling, however its exact function remains unclear. To use a protein like Ecd to alter glycolysis is in my opinion not ideal, because it is not known what EcdIR is causing at a mechanistic level and since glycolysis is very well characterized the authors could have chosen many other ways to affect it in a cleaner way.

We acknowledge that Ecd may not have been the most obvious choice for a method to perturb cell metabolism in the wing disc. This choice was for historical reasons, which we did not sufficiently explain in the manuscript. However, Ecd does have a known role in promoting glycolysis that is preserved from yeast to humans. Furthermore, our argument that glycolytic ATP affects Smo accumulation is not only based on the Ecd data, but also on more direct perturbation of Gapdh and Glo1 in wing imaginal discs. We also present data from mammalian cells showing that pharmacological inhibition of glycolysis (at two different steps) induces Smo ciliary translocation. Taken together, the data support the argument that changes in glycolysis resulting in lower ATP levels activate Smo in a way that is conserved across phyla.

To further strengthen the connection between glycolysis, Ecd, and the Hh pathway, we have made the following additional changes to the manuscript:

1. We added a paragraph to the introduction section explaining how we became interested in Ecd and why we thought this protein could be used to address the effect of glycolysis on Hh signaling (p4-5, lines 91-101).
2. We added a whole new section to the results (p6-7, lines 114-152), coupled with a new figure (now Fig 1), showing how steady state levels of ATP are lower upon RNAi-mediated knock-down of three glycolytic enzymes: Pfk, Gapdh, and Glo1.
3. We added supplemental data (Appendix Fig S2) and corresponding text in the results section (p7, lines 145-152) demonstrating that RNAi of glycolytic enzymes can affect wing shape by making them rounder. These data suggest that affecting glycolysis not only affects the amount of growth but its direction and/or proportions, which are controlled by the patterning systems, including Hh.
4. We added new data showing that perturbation of Ecd also similarly affects adult wing shape (Fig 2, p9, lines 190-198). Together with the data on ATP levels, these data indicate that loss of Ecd phenocopies loss of glycolytic enzymes such as Gapdh. This phenotype is consistent with the observation that Hh pathway is ectopically activated – loss of glycolysis impedes growth along the PD axis, but upregulated Hh signaling maintains growth along the AP axis.
5. We summarize the argument that Ecd's function in promoting glycolysis is conserved in *Drosophila* at the end of the second section of the results, p9, lines 199-206.

As previously reported for Ecd in the PG, the authors now find that Ecd in the wing disc somehow regulates splicing and that it specifically affects splicing of Gapdh, Pdk and PyK. This defect could serve as an explanation for how Ecd affects glycolysis levels. It remains unanswered if these splicing

defects indeed lead to a decrease in glycolysis (for instance with measurement of lactate), or if Ecd also affects splicing of other metabolic enzymes (for instance in Krebs cycle). The authors also directly try to affect glycolysis by knocking down Gapdh2 (again no lactate direct measurement) and find that both Smo and Ci155 are affected in a similar manner as in EcdIR.

We have not exhaustively characterized the splicing changes in *ecd*^{RNAi} tissue, although this would be interesting. While we cannot rule out the fact that Ecd may affect the splicing of non-glycolytic targets, our data are consistent with the hypothesis that the phenotypes of reduced ATP levels, upregulation of Hh signaling, and altered adult wing morphology are due to changes in splicing of glycolytic enzymes upon loss of Ecd. We have tried to address whether the *ecd*^{RNAi} phenotype can be reversed by expressing normally spliced versions of glycolytic enzymes. No single enzyme was able to do this, and the genetic combinations required to express all of them proved too cumbersome. If loss of Ecd affects multiple glycolytic enzymes, it isn't surprising that restoring one of them doesn't reverse the Ecd phenotype. We did not include this inconclusive data in the manuscript.

Quantifying lactate levels is a good suggestion but would require additional assay development, which would further delay the manuscript. Furthermore, it is not entirely clear that lactate production is a perfect readout of glycolysis activity in the wing disc. Fig 1 shows that ATP levels drop significantly upon inhibition of oxidative phosphorylation, indicating that this pathway is active, and thus the wing disc is not likely to be undergoing aerobic glycolysis or a "Warburg effect", as in some cancer cells. Thus, the carbon units coming from glycolysis may be funneled through the Krebs cycle and into OxPhos rather than being secreted as lactate. Nonetheless, we do show phenotypic consequences of the loss of glycolytic enzymes: lower steady state ATP levels, upregulated Hh, and misproportioned adult wings. While loss of any one enzyme in the pathway may not be sufficient to reduce glycolysis, we have now included considerable additional supporting data where we perturb not just Gapdh, but other enzymes, including Pfk, which is a key point of control in the pathway.

The authors analysis of the role of Hh release in the accumulation of Smo and Ci155 in the background of EcdIR or GapdhIR (in Dispatched mutant background) is not very convincing since the increase in both Smo and Ci155 in these backgrounds is very small and may not be significant.

We performed statistical tests and found indeed that the differences are not significant. We have added this information to the figure legend and modified our discussion of this result in the text (pg 11, lines 249-258).

Note, however, that we also show that there is no effect on Smo or Ci₁₅₅ when *ecd*^{RNAi} is confined to the posterior cells, which produce Hh (Appendix Fig S5, with added quantification). Furthermore, we have knocked down *ecd* in a Hh temperature-sensitive mutant and find that Ci₁₅₅ is elevated, even though target gene activation (*dpp*) is lost (Figure I below). We can include these data in the manuscript if you feel it is necessary.

Thus, in total, our data are consistent with the claim that Ecd does not absolutely require Hh ligand to affect the pathway. We acknowledge, however, that the Hh ligand-induced signaling contributes to the phenotype of Ecd. We have added such a statement to the results on pg 12, lines 271-272.

Figure I. Stabilization of Ci₁₅₅ upon loss of Ecd does not require the Hh ligand, but target gene expression does

A-A'' Time-controlled knock-down of *ecd* in the dorsal compartment of *hh* mutant wing discs (*hh^{ts2}, apGal^{ts}>ecd^{RNAi}*, see Fig 1C for the expression pattern of *apGal^{ts}*). IF of *hh^{ts2}, apGal^{ts}>ecd^{RNAi}* wing discs, stained for Hh (A), Ci₁₅₅ (A'), and *dppZ* (A''). Next to the image of Ci₁₅₅ stainings are

quantifications of Ci₁₅₅ staining intensity (n=2). Graphs show mean (thick line) ± SD (thin lines). Dashed yellow lines indicate the position of the A/P boundary. Loss of Ecd increases Ci₁₅₅ stabilization even in the absence of Hh. Wing discs were analyzed after 48 h of RNAi induction. Scale bars= 50 μm.

The decrease in plasma membrane potential and decreased accumulation of PAC-NAE in both EcdIR or GapdhIR are very interesting observations. It would be important to confirm if this is specific for Gapdh or if affecting glycolysis in other ways (or other metabolic pathways) also causes this same phenotype.

We have now confirmed that loss of membrane potential occurs upon knock-down of *Pfk*, which also affects Hh signaling, ATP levels, and wing shape (Fig EV5). Interestingly, not all enzymes of glycolysis had this effect. For example, we found that loss of Enolase does not cause a drop in membrane potential, nor does it affect Hh signaling, ATP levels or wing shape in the same way as the other glycolytic enzymes. To confirm that the RNAi was effective, we acquired an antibody from another lab and found that indeed the protein levels were significantly reduced. For other enzymes in which we saw no effect, however, it is difficult to conclude whether the RNAi was functional. We do not think these additional data significantly change or add to the story of the manuscript, so we have not included them.

It would also be important to test if disrupting membrane potential also leads to an accumulation of Smo and Ci155 and therefore this is the mechanism by which reduction in Gapdh (and potentially glycolysis levels) is interfering with the distribution/levels of these proteins.

We agree this is an interesting experiment. Interestingly, Ishwar Hariharan's lab very recently posted a preprint to the BioRxiv supporting our claim that altering membrane potential affects Hh signaling (Emmons-Bell et al 2020). They use RNAi to target two ion channels in the wing disc and find that membrane depolarization leads to increased Smo membrane accumulation, whereas hyperpolarization decreases Smo membrane accumulation. They confirm these results with different tools expressed in the salivary glands: overexpression of a bacterial channel, NaChBac, induces membrane depolarization and increases Smo membrane accumulation and Ptc expression; conversely, overexpression of Kir2.1 induced hyperpolarization and decreased Smo membrane accumulation. Lastly, they optogenetically altered membrane potential in the salivary gland and show a rapid (~10min) response of Smo membrane accumulation in response to altering membrane potential.

While these new data support our work, no mechanism was proposed for how membrane potential affects Hh signaling. Our work showing that membrane perturbation affects uptake of inhibitory lipids for the Hh pathway provides a connection linking glycolysis to membrane potential and Hh signaling.

Overall this manuscript addresses a very interesting question, it has a series of very interesting experiments/data but in the current format it is hard to conceptually integrate them all.

We hope that the addition of new data on glycolytic enzyme perturbation and the modification of the text of the introduction and results makes the connection between glycolysis, Ecd, and Hh signaling more apparent.

1- 92 - even if Ecd has been shown to not be involved in steroid hormone production it could still be involved in ecdysone signaling.

Our point here was only to address the previous literature showing that Ecd is involved in steroid production. A role for Ecd in mediating ecdysone signaling is not established, nor is there evidence suggesting that ecdysone signaling affects glycolytic enzymes. If Ecd were required autonomously in the wing disc for ecdysone *signaling*, we may expect to see more significant and broader effects patterning. In particular, we would expect a defect in Wg expression or signaling (Dye et al 2017, Mitchell et al 2013). We found no changes to the expression of Wg or its target Cut upon loss of Ecd (Figure II below), however. We can add these data to the manuscript if requested, but we feel doing so may distract from the main storyline of the paper.

Figure II. Loss of Ecd does not affect the expression of Wingless or Cut

A-D Time-controlled knock-down of *ecd* in the dorsal compartment of the wing discs (see Fig 1C for the expression pattern of *apGal^{ts}*). IF of control (A, C) and *apGal^{ts}>ecd^{RNAi}* (B, D) wing discs, stained for Wingless (Wg, A, B) or Cut (C, D). Next to the images are quantifications of the respective stainings along the D/V boundary of control and *apGal^{ts}>ecd^{RNAi}* wing discs (n=4 (C, D), n=5 (A, B)). Graphs show mean (thick line) \pm SD (thin lines). Dashed green lines indicate the position of the D/V boundary. To compare stainings between control and *apGal^{ts}>ecd^{RNAi}* wing discs, pixel intensities in control wing discs were adjusted to match that of *apGal^{ts}>ecd^{RNAi}* wing discs in the ventral compartment (mean \pm SD as grey lines in B and D). Statistical analyses (t-test) revealed no significant difference in Wg or Cut expression in the dorsal compartment between control and *apGal^{ts}>ecd^{RNAi}* wing discs. Thus, loss of Ecd does not affect Wg or Cut expression. Wing discs were analyzed 48 h after RNAi induction. Scale bars= 50 μ m.

2- Fig. 1B' and C' - Caspase 3 quantification missing.

These data have been moved to Fig EV1. Quantification of activated Caspase-3 staining has now been added in part H.

3- Fig. EV1 - Is Ecd a regulator of splicing? Would you see the same phenotype if you would knocked down a bone-fide splicing regulator (e.g Nelf-a)?

Yes – we have done this experiment and see that knock-down of *brr2* (a splicing factor that interacts with Ecd) has the same effect on *Gapdh* splicing and *Smo* accumulation as *ecd^{RNAi}*. We now include this data in Appendix Fig S4A, A' and Fig EV4C-C''.

4- Fig. EV1 - Is this role of Ecd in splicing specific for the enzymes tested? Did you look at other glycolysis enzymes or to Krebs cycle enzymes? If Ecd affects splicing of multiple enzymes in multiple pathways the overall effect in metabolism can be unpredictable.

We have not exhaustively characterized the splicing changes in *ecd^{RNAi}* tissue. We show, however, that the phenotype with respect to ATP levels and adult wing shape is similar between Ecd and glycolytic enzymes such as *Gapdh*. If Ecd had many other functions or splicing targets, we may expect a different phenotype. Furthermore, there is already a known function for Ecd in promoting glycolysis that is conserved from yeast to humans.

5- Lower ATP levels do not necessarily mean lower glycolysis; lower levels of spliced *Gapdh*, *Pyk*, etc do not mean lower glycolysis - Quantify glycolysis (e.g. lactate chemical measurement).

As explained above, we do not agree that lactate measurements would necessarily be an effective readout of glycolysis in the wing disc. What we have done instead is to expand our analysis of ATP levels and adult wing phenotypes to many glycolytic enzymes, including a key point of control in glycolysis, *Pfk*. The conservation of Ecd function from yeast to humans strongly suggests that it has the same function in the wing disc, although we agree that more experiments are needed as

direct proof. Nonetheless, the fact that *ecd*^{RNAi} alters splicing of glycolytic enzymes and phenocopies the loss of glycolytic enzymes is consistent with this idea.

6- The authors mention that Ecd mutants can be rescued when UAS-Ecd x Ptc-Gal4 - Could Ecd be also required for the splicing of Dpp? The experiment with dpp-LacZ does not answer this question.

This is an interesting question but outside the scope of the paper. There is no evidence that Dpp signaling feeds back on Hh signaling in the wing disc, so such a mechanism would be unlikely to explain the effect of loss of Ecd on Smo, which is the focus of this paper. The key insight from the *ecd*^{RNAi} experiment is that it lowers steady state ATP levels, lowers the plasma membrane potential, and reduces N-acylethanolamide uptake. We think that these are the key perturbations that result in Smo destabilization, because all of these effects are mimicked by RNAi targeting the glycolytic enzymes *Gapdh* and *Pfk*, which also stabilizes Smo.

7- 154 - "glycolysis on Smo trafficking" the effect on trafficking was not really shown in Drosophila. The accumulation of Smo could have many causes. Are total levels of Smo in Drosophila EcdIR/Gapdh and in mammalian cells treated with 2-DG the same?

We have changed "trafficking" to "membrane accumulation" in all instances.

8- 165 - "trafficking" - not shown.

See point 7.

9- Fig 3C-F - in the absence of Disp, loss of Ecd does not cause the same accumulation of Smo or Ci as in the presence of Disp. This difference could indeed mean that Hh release is required. How can this decrease be otherwise explained?

See explanation above – we have modified the text to claim that Hh is not required although ligand-induced signaling can contribute to the phenotype of loss of Ecd.

10- 183 - Lpp = meaning? HhN=meaning?

We have added clarification to pg 11, line 260: lipoprotein-associated Hh (Lpp-associated) and sterol-free Hh (HhN). We have also added a description of these abbreviations in the Introduction section (p4, lines 84-89).

11- Fig. EV1 H' - Show area of gel which would show the unspliced form of eIF-4a and rp49 (as in G' or F').

We have now included spliced and unspliced mRNA of *eIF-4a*. These data have been moved to Appendix Fig S3 E-E'. We did not have the unspliced available for *rp49*, so we have removed it.

12- Fig. EV1 H' - Why is there no band in gel in DNA lane for eIF4a? Why is the band of rp49 in DNA lane heavier?

These data now appear in Appendix Fig S3 E'. The band in the DNA lane was simply not in the cropped region. We now include a cropped image that also shows the higher running DNA band for *eIF-4a*. Data for *rp49* was removed (see point 11).

Minor comments:

Title: It is too general. A title more directed to the results of the manuscript would be better.

We have changed to: "Glycolysis regulates Hedgehog signaling via the plasma membrane potential"

Abstract: Line 25 - "We show that reducing glycolysis" - the authors have never shown this. Reducing the levels of ATP can have many causes. It is also not clear if the tools the authors use indeed lead to a reduction in glycolysis.

We have added data showing that knock-down of glycolytic enzyme expression reduces ATP levels, similar to Ecd knockdown. This statement is also supported by the pharmacological inhibition of glycolysis in cell culture.

Introduction: Line 42 - remove "respectively". A-KG could both affect methylation and acetylation.
Done.

Line 53 - "Smo can be activated by cholesterol..." in vertebrates.
"in vertebrates" has been added (now pg 3, line 67)

Referee #3:

Major comments:

1. First, my enthusiasm for this study was diminished by the very small size of the effects in nearly every case (Fig. 1E & G, 2B-E, 3D & F, and 4A). Although these differences are shown to be statistically significant (in the case of Fig. 1 and 4), it is not clear that these differences are biologically meaningful. The authors do show that one biologically relevant output, dpp expression, is affected (Fig. 3H & J), but additional evidence along these lines would be helpful. Alternatively (or in addition), it would be interesting to determine whether differences in Ecd expression and/or the rate of glycolysis in contributes to the normal patterning of Hh signaling in wildtype tissue in a meaningful way. A third possibility would be to assay for more dramatic effects on Hh signaling or biological outcome in homozygous mutant clones. Although it may not be necessary to pursue all three of these lines of inquiry, I think that some additional data about either the biological impact of the experimental perturbations that have already been used in this study or the ways in which changes in metabolism within wildtype tissue contributes to the patterning of Hh signaling is important and would significantly enhance the impact of the study.

1. New data regarding biological impact of experimental perturbations: We now include data showing that adult wing size and shape changes after uniform knock-down of several glycolytic enzymes (including *Gapdh*) in the wing – the wings are smaller and misproportioned (shorter along the PD axis and broader along the AP axis, Appendix Fig S2). Finally, we have constructed a dominant negative version of Ecd whose expression reproduces the effects of *ecd^{RNAi}* on *Gapdh* splicing and Smo stabilization without inducing the cell death that occurs after long-term *ecd^{RNAi}*. While driving *ecd^{RNAi}* throughout the wing results in only vestigial wings, expressing the dominant negative construct gives rise to wings that look like *Gapdh^{RNAi}* wings (Fig 2).

It is true that loss of glycolysis has relatively weak effects. This finding is consistent with the fact that we see a huge effect on ATP levels of inhibiting oxidative phosphorylation (Fig 1), indicating that the developing wing does not rely solely on glycolysis for its energy, as some other cell types. This finding is interesting on its own, as it contrasts the wing disc with other tumor models that exhibit aerobic glycolysis. Nonetheless, the phenotypes that we see are significant, indicating that glycolysis does have a function in promoting wing disc growth and pattern, ultimately contributing to final wing shape and size.

2. How does metabolism contribute to patterning of Hh signaling? This very interesting question is the subject of ongoing work in the lab. Of course, nutrition may cause metabolic changes in the wing disc through the insulin pathway, and we have a student in the lab who is characterizing such effects and how they may influence Hh activity. We are also investigating how Hh signaling may influence metabolism in the wing disc, as it does in many tumors. These projects are yet incomplete, however. Furthermore, including them would significantly lengthen the manuscript.

3. More dramatic effects in homozygous mutant clones. Unfortunately, homozygous *ecd* mutant clones are excluded from the wing disc (Gaziova et al 2004), likely due to cell competition. Thus, interpretation of this kind of experiment would be complicated.

2. In Figures 2 and 3, the standard deviations of the red and black lines overlap in most cases, raising the possibility that the differences between the means are not statistically significant. A statistical procedure to should be used to test whether these data sets are significantly different from each other.

We have now performed tests of significance for all data. We report the p-values in the figures legends and describe the test in the methods section. We have also revised the section of the results (pg 11, lines 249-258) explaining the old Fig 3 (now Fig 5).

3. In Fig. 3I, the relevant parts of the image (the primary cilia) are located around the periphery of the image and are almost obscured by the image label text. An alternate image or layout should be used so that it is easier to see the effect.

Data for this experiment are now in Fig 4. We have added close up images to improve visualization.

DESCRIPTION OF CHANGES TO FIGURES:

- In response to reviewer requests, we have added new data and reorganized the main figures, as well as the supplement. To accommodate these new data, while staying within the length restrictions, we now have an appendix with additional supplemental data. We now have 6 main figures, 5 supplemental figures (EV), and 6 appendix figures. Specific changes to the figures are outlined in detail below.

Fig. 1 and associated supplement:

- NEW DATA added to Fig 1, showing reduction of FRET-ATP upon knock-down of glycolytic enzymes (*Pfk*, *Gapdh*, *Glo1*). Note we have improved the visualization of the data, showing line graphs connecting the dorsal and ventral compartments of each analyzed wing disc, rather than presenting bar graphs of pooled data. We also slightly modified the cartoons.
- OLD Fig 1A-D'' showing the growth phenotype upon loss of *Ecd* was moved to Fig EV1D-G.
- Fig EV1 includes NEW DATA showing that *apGal^{ts}* does not express in the ring gland, and thus ecdysone production should be unaffected by *apGal^{ts}>ecd^{RNAi}*.
- OLD Fig EV1 was shifted down and split into two: OLD D-G'' is now in Fig EV2; OLD A-C and H-H'' is now in Appendix Fig S3.
- NEW DATA added to Appendix Fig S1, showing a confirmation that *Gapdh^{RNAi}* affects both *Gapdh1* and *Gapdh2*.

Fig. 2 and associated supplement:

- Fig 2 is now entirely NEW DATA, combined with NEW Appendix Fig S2. In these figures, we show the consequences of loss of glycolytic enzymes and *Ecd* on adult wing morphology.
- OLD Fig 2 is now split into two subsequent figures: A-E' is now in Fig 3; F-K is now in Fig 4.
- NEW DATA added to Appendix Fig S4, showing that RNAi of *brr2*, a component of the core splicing machinery, also affects *Gapdh2* splicing.

Fig. 3 and associated supplement:

- NEW DATA added to Fig 3, showing that knock-down of *Pfk* (C-C') has the same effect on the Hh pathway as that of *ecd*, *Gapdh*, and *Glo1*.
- OLD Fig EV2 is unchanged but is now labeled Fig EV3.
- NEW DATA added to Fig EV4, showing that over-expression of dominant negative *Ecd* (*ecd^{DN}*) and *brr2* RNAi have similar effects on *Ci₁₅₅* and *Smo* as RNAi of *ecd* and glycolytic enzymes.

Fig. 4 and associated supplement:

- Fig 4 is the same data from OLD Fig 2F-K. We added magnified images for better visualization.

Fig. 5 and associated supplement:

- OLD Fig. 3, reordered so that Dpp expression +/- Dispatched comes before *Smo/Ci₁₅₅*.
- OLD Fig EV3 is now Appendix Fig S5 and S6.

Fig. 6:

- OLD Fig 4, unchanged.
- NEW DATA added to Fig EV5, showing change in membrane potential upon loss of *Pfk*.

Dear Natalie, dear Stephanie,

Thank you for submitting your revised manuscript for consideration by The EMBO Journal. My apologies for the extended duration of the re-review of your manuscript due to protracted referee input. Your revised study was sent back to the three referees for evaluation, and we have received comments from all of them, which I enclose below. As you will see the referees find that their concerns have been sufficiently addressed and they are now broadly in favour of publication.

Thus, we are pleased to inform you that your manuscript has been accepted in principle for publication in The EMBO Journal, pending the minor remaining issues and additional aspects related to formatting and data representation as detailed below are addressed at re-submission.

Please contact me at any time if you have further questions related to below points.

Thank you for giving us the chance to consider your manuscript for The EMBO Journal. I look forward to your final revision.

Again, please contact me at any time if you need any help or have further questions.

Kind regards,

Daniel

Daniel Klimmeck PhD
Editor
The EMBO Journal

>> Please check and correct figure callouts: Figure 2 is called out before Figure 1 L,M; sub-panels of Figure 4 are not called out.

>> Provide main figures and EV figures as individual, high-resolution figure files.

>> Indicate re-use of Figure EV2B eIF-4a in Figure S3E in the figure legend.

>> Indicate the microscopy tiling approach applied in figure legend 2D.

>> Provide an ORCID for all corresponding authors (S.S.)

>> Please consider additional changes and comments from our production team as indicated by attached .doc file and leave changes in track mode.

The revision must be submitted online within 90 days; please click on the link below to submit the revision online before 12th Oct 2020.

Link Not Available

Referee #1:

The authors have addressed my concerns which mostly centered on the small differences that were quantified but was hard to see in the IF, and whether Ecd is affecting growth in general via non-wing dependent mechanisms. I feel the data was strengthened by the adult wing data, and confirmation with the use of the dominant negative form of Ecd. I am happy with the revisions, and feel that the paper is acceptable for publication in its current state.

Referee #2:

The authors have addressed all my points satisfactorily. I thank the authors for the additional figures in their rebuttal letter addressing some of my questions which were very helpful. I do not require these figures to be included in the manuscript. I think the manuscript is much improved and I am strongly in favor of acceptance and rapid publication.

Referee #3:

With this revised manuscript, the authors have addressed my concerns and I now fully support publication.

The authors performed the requested editorial changes.

Dear Natalie, dear Stephanie,

Thank you for submitting the revised version of your manuscript. I have now evaluated your amended manuscript and concluded that the remaining minor concerns have been sufficiently addressed.

Thus, I am pleased to inform you that your manuscript has been accepted for publication in the EMBO Journal.

Please note that it is EMBO Journal policy for the transcript of the editorial process (containing referee reports and your response letter) to be published as an online supplement to each paper. I would thus like to ask for your consent on keeping the additional referee figures included in this file.

Also in case you might NOT want the transparent process file published at all, you will also need to inform us via email immediately. More information is available here:

http://emboj.embopress.org/about#Transparent_Process

Please note that in order to be able to start the production process, our publisher will need and contact you regarding the following forms:

- PAGE CHARGE AUTHORISATION (For Articles and Resources)

[http://onlinelibrary.wiley.com/journal/10.1002/\(ISSN\)1460-2075/homepage/tej_apc.pdf](http://onlinelibrary.wiley.com/journal/10.1002/(ISSN)1460-2075/homepage/tej_apc.pdf)

- LICENCE TO PUBLISH (for non-Open Access)

Your article cannot be published until the publisher has received the appropriate signed license agreement. Once your article has been received by Wiley for production you will receive an email from Wiley's Author Services system, which will ask you to log in and will present them with the appropriate license for completion.

- LICENCE TO PUBLISH for OPEN ACCESS papers

Authors of accepted peer-reviewed original research articles may choose to pay a fee in order for their published article to be made freely accessible to all online immediately upon publication. The EMBO Open fee is fixed at \$5,200 (+ VAT where applicable).

We offer two licenses for Open Access papers, CC-BY and CC-BY-NC-ND.

For more information on these licenses, please visit: <http://creativecommons.org/licenses/by/3.0/> and http://creativecommons.org/licenses/by-nc-nd/3.0/deed.en_US

- PAYMENT FOR OPEN ACCESS papers

You also need to complete our payment system for Open Access articles. Please follow this link and select EMBO Journal from the drop down list and then complete the payment process:

https://authorservices.wiley.com/bauthor/onlineopen_order.asp

Notably, please be reminded that under the DEAL agreement of German scientific institutions with our publisher Wiley, you could be eligible for free publication of your article in the open access format. Please contact either the administration at your institution or Wiley (embojournal@wiley.com) to clarify further questions.

Should you be planning a Press Release on your article, please get in contact with embojournal@wiley.com as early as possible, in order to coordinate publication and release dates.

On a different note, I would like to alert you that EMBO Press is currently developing a new format for a video-synopsis of work published with us, which essentially is a short, author-generated film explaining the core findings in hand drawings, and, as we believe, can be very useful to increase visibility of the work.

Please see the following link for representative recent examples:
https://www.embopress.org/video_synopses

If you have any questions, please do not hesitate to call or email the Editorial Office.

Thank you again for this contribution to The EMBO Journal and congratulations on a successful publication. Please consider us again in the future for your work.

Kind regards,

Daniel

Daniel Klimmeck, PhD
Editor
The EMBO Journal
EMBO
Postfach 1022-40
Meyrhofstrasse 1
D-69117 Heidelberg
contact@embojournal.org
Submit at: <http://emboj.msubmit.net>

Corresponding Author Name: Stephanie Spann and Natalie A. Dye

Journal Submitted to: The EMBO J

Manuscript Number: EMBOJ-2019-101767